# ANNEALED IMPLICIT Q-LEARNING IN ONLINE REINFORCEMENT LEARNING

## ABSTRACT

In continuous action online reinforcement learning, actor-critic methods are predominantly used. However, compared to Q-learning-based discrete action algorithms that model the optimal Q-value, continuous action algorithms that model the Q-value for the current policy and perform policy improvement solely through policy updates suffer from low sample efficiency. This study investigates whether an algorithm that implicitly estimates the optimal Q-value, typically used in offline RL, is also effective in online RL. It is demonstrated that a loss function aimed at achieving optimality distorts the distribution of Q-values, leading to overestimation bias, and that this distortion and bias increase as learning progresses. To address this issue, we propose a simple algorithm that anneals optimality. Our method significantly outperforms widely used methods such as SAC and TD3 in online DM Control tasks. Additionally, we demonstrate that annealing improves performance and enhances robustness to the hyperparameter related to the optimality.

## 1 INTRODUCTION

In recent years, online reinforcement learning (RL) has been widely applied in fields such as robotics (Schulman et al., 2015; Haarnoja et al., 2018) and games (Mnih et al., 2013; 2015; Silver et al., 2016), particularly demonstrating remarkable performance in discrete-action tasks. In Q-learning-based algorithms, commonly used for discrete-action tasks, the optimal value is estimated through optimal Bellman operator, where the target value is calculated as $r + \gamma \max_{a'} Q(s', a')$. However, in continuous-action tasks, it is challenging to compute $\max_{a'} Q(s', a')$ over an infinite number of actions, so actor-critic-based algorithms estimate the Q-value through Bellman backups based on the current policy, where the target value is $r + \gamma \mathbb{E}_{a' \sim \pi} Q(s', a')$. In these cases, policy improvement is achieved solely through policy updates, resulting in slower performance improvements and reduced sample efficiency. This is a significant issue for continuous-action tasks, such as robot control, which are highly complex and require sample efficiency due to data collection difficulties. While previous studies have proposed methods for modeling the (soft) optimal value, they have encountered issues such as increased computational complexity and learning instability due to the exponential terms involved (Haarnoja et al., 2017; Kalashnikov et al., 2018; Garg et al., 2023).

This study examines the effectiveness of modeling optimal value in online RL. To model the optimal value, we employ the expectile loss (Kostrikov et al., 2022), which is typically used in offline RL and implicitly models the maximum value. Expectile loss is a stable loss function that allows control over the degree of optimality via the hyperparameter $\tau$. As $\tau$ approaches 1, the function behaves similarly to a maximum operator.

First, this study examines the increase in Q-value skewness and the accumulation of overestimation bias resulting from estimating the optimal value. Using expectile loss, an asymmetric loss function, for estimation may lead to skewness in the Q-value distribution. We explore how this skewness influences bias. While Thrun & Schwartz (1993) demonstrated the impact of distribution variance and the number of actions on bias after taking the max, our study numerically shows that the skewness of the original Q-value distribution can also affect bias. In other words, as optimality increases, the skewness of the Q-value distribution grows, leading to significant bias, which accumulates and potentially destabilizes the learning process toward the end.

Thus, while pursuing optimality carries the risk of performance degradation, a certain degree of bias is beneficial during the early stages of learning as it encourages exploration (Lan et al., 2020). It is reasonable to pursue optimality during the early phases before significant skewness and bias have accumulated. Toward the end of the learning process, the current policy approaches the (sub)-optimal policy, and the Q-value nears the (sub)-optimal Q-value, making stable estimation more desirable than highly optimal estimation.

In this context, we propose annealing the optimality in Bellman backups by gradually shifting from a Q-learning-based target $r + \gamma \max_{a'} Q(s', a')$ to a SARSA-based target $r + \gamma \mathbb{E}_{a' \sim \pi} Q(s', a')$. By using expectile loss, we smoothly interpolate between these two estimations. By annealing the parameter corresponding to the expectile being estimated, we can tolerate bias and pursue optimality through Q-learning-based updates during the exploration-heavy early learning stages. In the later stages, when the policy is close to (sub)-optimal, we transition to SARSA-based updates to prevent further accumulation of bias.

We conducted experiments using continuous-action online RL tasks in DM Control, combining the proposed annealed Implicit Q-learning (AIQL) with SAC. Compared to widely-used algorithms like SAC and TD3, AIQL demonstrated significantly superior performance. Additionally, when compared to static optimality estimation, annealing was shown to enhance robustness to the optimality hyperparameter and improve performance.

The contributions of this study are as follows:

- We investigated and experimentally tested the modeling of optimal Q-value using implicit Q-learning loss in continuous-action online RL.
- Through numerical experiments, we confirmed that modeling optimal Q-value using implicit Q-learning loss can amplify bias due to skewness in the Q-value distribution.
- We proposed optimality annealing for the value function modeling, based on the assumption that early bias is beneficial for exploration and the Q-value approaches optimality in the later stages of learning. Experimental results demonstrated that annealing improves both robustness to the hyperparameter and performance.

## 2 PRELIMINARIES

### 2.1 REINFORCEMENT LEARNING

In RL, the problem is defined within a Markov decision process (MDP) framework presented by the tuple $(\mathcal{S}, \mathcal{A}, \mathcal{P}, r, \gamma, d)$. Here, $\mathcal{S}$ denotes the set of all possible states, $\mathcal{A}$ denotes the set of all possible actions, $\mathcal{P}(s_{t+1}|s_t, a_t)$ is the transition probability from one state to another given a specific action, $r(s, a)$ represents the reward function assigning values to each state-action pair, $\gamma$ is the discount factor that diminishes the value of future rewards, and $d(s_0)$ is the probability distribution of initial states. The policy $\pi(a \mid s)$ is the probability of taking a specific action in a given state. The objective of RL is to discover a policy that maximizes the expected sum of discounted rewards, denoted as $\mathbb{E}[R_0 \mid \pi]$, where $R_t$ is the return calculated as $R_t = \sum_{k=t}^{T} \gamma^{k-t} r(s_k, a_k)$ and $T$ is a task horizon.

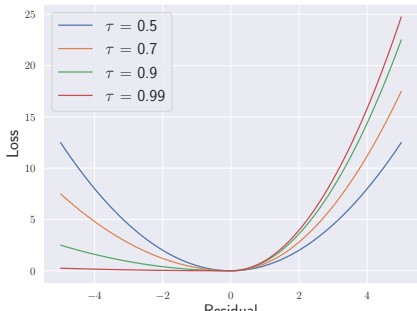

Figure 1: Expectile loss used in IQL.

### 2.2 IMPLICIT Q-LEARNING

Implicit Q-learning (IQL) is an algorithm widely used in offline RL to learn the value function and the policy using only actions from the offline dataset, without relying on actions from the policy being learned. By avoiding the use of out-of-distribution actions in learning, the problem of distribution shift in offline RL can be mitigated. IQL can also learn near-optimal value functions in

an in-distribution manner. This is achieved by considering the target value of the value function as a random variable depending on the action and estimating the upper expectile. Thus, the loss for learning the Q-function is as follows:

$$L(\theta) = \mathbb{E}_{(s,a,s',a')\sim\mathcal{D}}[L_2^\tau(r(s,a) + \gamma Q_{\bar{\theta}}(s',a') - Q_\theta(s,a))],$$

where $L_2^\tau(u) = |\tau - \mathbb{1}(u < 0)|u^2$, $Q_\theta$ is a parameterized Q-function, and $Q_{\bar{\theta}}$ is the target network.

This loss function is asymmetric as shown in Figure 1, and it is equivalent to the L2 loss when $\tau = 0.5$. As $\tau$ approaches 1, the loss for negative errors decreases, leading to the estimation of larger values. In IQL, near-optimal values are estimated by using $\tau$ close to 1, e.g., $\tau = 0.9$. In a stochastic environment, not only the randomness of the action but also the randomness of the state transition affects the target value, so the following loss function involving the V-function is used:

$$L(\psi) = \mathbb{E}_{(s,a)\sim\mathcal{D}}[L_2^\tau(Q_{\bar{\theta}}(s,a) - V_\psi(s))],$$
$$L(\theta) = \mathbb{E}_{(s,a,s')\sim\mathcal{D}}[(r(s,a) + \gamma V_\psi(s') - Q_\theta(s,a))^2].$$

Thus, in IQL, only the actions from the dataset were used to compute the maximum in the optimal Bellman equation.

# 3 ANNEALED IMPLICIT Q-LEARNING

## 3.1 IQL IN ONLINE RL

Algorithms for online RL in continuous action tasks, such as SAC and TD3, estimate the value function for the current policy and rely on policy updates to improve the policy. This study aims to enhance sample efficiency by estimating the optimal value during value function learning. In IQL, expectile loss is used to compute the optimal value based on the policy used during dataset creation. This method is effective for computing the maximum in continuous action tasks. In this study, we use expectile loss in online RL to compute the optimal value based on the actions of the current policy. The loss for the value function is defined as follows:

$$L(\theta) = \mathbb{E}_{(s,a,s')\sim\mathcal{D},a'\sim\pi}[L_2^\tau(r(s,a) + \gamma Q_{\bar{\theta}}(s',a') - Q_\theta(s,a))].$$

In online RL, where actions are sampled from the current policy to evaluate Q-values, the issue of overestimation bias becomes more pronounced. The next section discusses the problem of overestimation bias when estimating near-optimal values.

## 3.2 OVERESTIMATION BIAS

Previous studies on online RL have highlighted that the overestimation of action values poses a significant challenge to the performance of RL algorithms. This bias is caused by noisy estimates that occur when using function approximators in value function estimation, as formulated by Thrun & Schwartz (1993). Following Thrun & Schwartz (1993), we assume that the estimated $\hat{Q}$ has i.i.d. noise $\epsilon$ added to the target $\bar{Q}$:

$$\hat{Q}(s,a) = \bar{Q}(s,a) + \epsilon_{s,a}.$$

Furthermore, following Thrun & Schwartz (1993), consider the case where all $\bar{Q}(s,a)$ values are equal across all actions, which represents the scenario where overestimation bias is the largest. In this case, the overestimation bias $Z_s$ is calculated as follows:

$$\begin{aligned} Z_s &= r(s,a) + \gamma \max_{a'} \hat{Q}(s',a') - (r(s,a) + \gamma \max_{a'} \bar{Q}(s',a')) \\ &= \gamma(\max_{a'} \hat{Q}(s',a') - \max_{a'} \bar{Q}(s',a')) \\ &= \gamma \max_{a'} \epsilon_{s',a'}. \end{aligned} \tag{1}$$

Thrun & Schwartz (1993) assumed $\epsilon$ followed a uniform distribution $U(-c,c)$, showing that the average overestimation $\mathbb{E}[Z_s]$ is $\gamma c \frac{n-1}{n+1}$, and it increases with the variance of noise $c$ and the number of actions $n$, even though the mean of $\epsilon$ is 0.

As mentioned above, it has been shown that overestimation bias arises from approximating the value function using a function approximator. However, in online RL, this bias should be partially corrected through interaction with the environment. Specifically, when the Q-value for a particular action increases, that action is more likely to be selected, and the bias is corrected using the observed rewards. However, if the bias becomes too large, the correction may not keep up, leading to its gradual accumulation and eventual failure of learning.

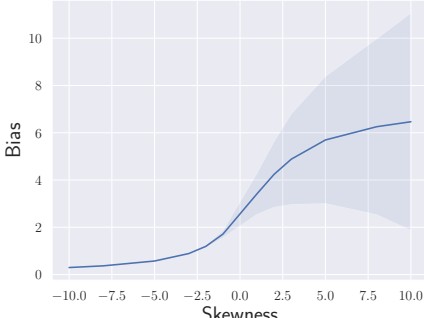

Figure 2: The bias $\mathbb{E}[Z_s]$ when $\epsilon$ follows an inverse Gaussian distribution with mean 0 and variance 1 in Equation (1) The details are provided in the appendix.

We consider the skewness in the distribution of Q-values as one of the causes for the occurrence of such excessive bias. Even if $\epsilon$ is symmetric, as assumed by Thrun & Schwartz (1993)'s uniform distribution, calculating the maximum results in a right-skewed distribution according to extreme value theory as discussed in Garg et al. (2023). This skewed noise propagates through the Bellman backup. Even when the maximum is calculated implicitly, such as in the case of expectile loss, the resulting estimates are likely to be skewed to the right due to the asymmetric nature of the loss function.

Therefore, we examine how bias changes when $\epsilon$ follows a skewed distribution instead of a uniform distribution. We use an inverse Gaussian distribution with parameters specifically designed to control skewness. The mean of the distribution is set to 0 by subtracting the location parameter after each sample is drawn. We calculate bias by averaging the maximum of $n$ noise samples, where $n$ corresponds to the number of actions. Figure 2 shows the bias magnitude with varying skewness, indicating that both the mean and variance of bias increase with greater skewness. This increase in bias variance can hinder learning, as suggested by Chen et al. (2021).

This experiment suggests that the increase in the skewness of the Q-values leads to an increase in both the mean and variance of the bias. As explained in Section 4.5, the skewness also grows throughout the learning process, and both skewness and bias become significantly large near the end of the learning process. Therefore, we address the accumulation of this skewness and bias.

### 3.3 Annealing of Optimality $\tau$

As discussed, overestimation bias arises from the calculation of $\max_{a'} Q(s', a')$. With expectile loss, as $\tau$ approaches 1, the computation increasingly resembles a max operation, while at $\tau = 0.5$, it calculates the expectation $\mathbb{E}_{a'}[Q(s', a')]$. In other words, overestimation bias increases as $\tau$ nears 1 and decreases as it approaches 0.5. At the same time, when $\tau$ is near 1, the computation approximates the optimal value; at $\tau = 0.5$, it represents the value function of the policy.

Thus, $\tau$ controls the trade-off between optimality and bias. Considering the impact of optimality and bias at different stages of learning, optimality, which leads to high sample efficiency, is crucial early in learning. A certain amount of bias promotes exploration (Lan et al., 2020), making a higher $\tau$ beneficial initially. Bias accumulates gradually, so it remains manageable early on. Therefore, a high $\tau$ that trades off some bias for optimality is preferable at the start.

In the later stages, accumulated bias needs to be minimized. As learning progresses, the value function and policy gradually converge, reducing the necessity for policy improvement. Hence, a smaller $\tau$ is favorable to control bias.

This study proposes setting $\tau$ close to 1 at the start and annealing it to 0.5 by the end of learning. At $\tau = 0.5$, the loss becomes an L2 loss, equivalent to estimating Q under the current policy. While our approach is not restricted to specific annealing strategies, our implementation linearly anneals $\tau$: given an initial $\tau$ value $\tau_{\text{init}}$ and a maximum number of timesteps $T$, $\tau$ at timestep $t$ is:

$$\tau(t) = \tau_{\text{init}} - (\tau_{\text{init}} - 0.5)\frac{\min(t, T)}{T}.$$

We name this method Annealed Implicit Q-learning (AIQL). The next section validates the effectiveness of this approach through experimental results.

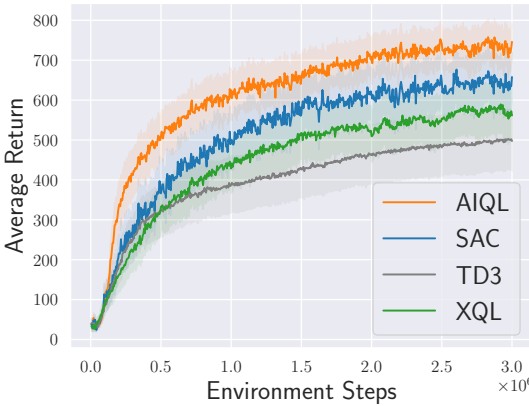

Figure 3: The average return across all tasks.

Table 1: The average score at 1M steps.

|  | Mean | IQM |
|---|---|---|
| AIQL | **624.8** $\pm$ 54.4 | **651.5** $\pm$ 41.9 |
| SAC | 493.7 $\pm$ 66.2 | 516.1 $\pm$ 56.7 |
| TD3 | 388.1 $\pm$ 69.8 | 339.6 $\pm$ 64.1 |
| XQL | 428.6 $\pm$ 62.9 | 423.8 $\pm$ 54.9 |

Table 2: The average score at 3M steps.

|  | Mean | IQM |
|---|---|---|
| AIQL | **746.1** $\pm$ 47.6 | **832.4** $\pm$ 17.2 |
| SAC | 657.9 $\pm$ 63.3 | 765.0 $\pm$ 35.5 |
| TD3 | 497.4 $\pm$ 75.3 | 526.5 $\pm$ 78.4 |
| XQL | 561.4 $\pm$ 64.8 | 623.8 $\pm$ 55.3 |

## 4 EXPERIMENTS

We conducted experiments using continuous action online RL tasks from the DM Control suite (Tassa et al., 2018; Tunyasuvunakool et al., 2020), integrating the proposed AIQL with SAC (Haarnoja et al., 2018).

To verify the effectiveness of the proposed method, we conducted several experiments, examining the following aspects:

- How the proposed Annealed IQL performs in comparison to widely-used online RL methods such as SAC and TD3.

- How expectile loss and $\tau$ annealing affect the skewness and bias of the Q-values.

- The performance variations based on different annealing methods. Specifically, we explore whether non-linear annealing methods are more effective compared to the linear method.

### 4.1 EXPERIMENTAL SETUP

We used 10 challenging continuous control tasks from DM Control as the experimental environment for online RL. As baseline methods, we employed SAC and TD3, widely used in continuous action tasks for online RL, along with XQL (Garg et al., 2023). XQL uses Gumbel regression, assuming a Gumbel distribution for the error distribution, and estimates the soft-optimal value in entropy-maximizing RL, offering a comparison as another method for estimating the optimal value.

Our proposed method, AIQL, was integrated with SAC. The only modification to SAC was the use of expectile loss instead of L2 loss for estimating the value function. The experiments that integrate TD3 and AIQL are described in appendix D. While Kostrikov et al. (2022) incorporated both a Q-function and a V-function to mitigate performance degradation observed in stochastic environments, AIQL demonstrated superior performance without employing a V-function, even in extremely stochastic environments. As a result, only the Q-function is utilized in our approach. Further details are provided in appendix F. The parameter $\tau$ was annealed linearly, starting at 0.9 at the beginning of the training and decreasing to 0.5 at 3 million steps. The results for using different annealing durations are presented in appendix E. Other hyperparameters of AIQL were kept the same as SAC, and at $\tau = 0.5$, AIQL is identical to SAC. The measurements of skewness and bias are based on Chen et al. (2021).

### 4.2 COMPARISON WITH PRIOR STUDIES

The average scores across all 10 tasks from DM Control for AIQL and the baseline methods from prior studies are shown in Figure 3. The average scores, interquartile mean (IQM) and their confidence intervals are listed in Table 1 and 2. AIQL significantly outperformed the baseline methods,

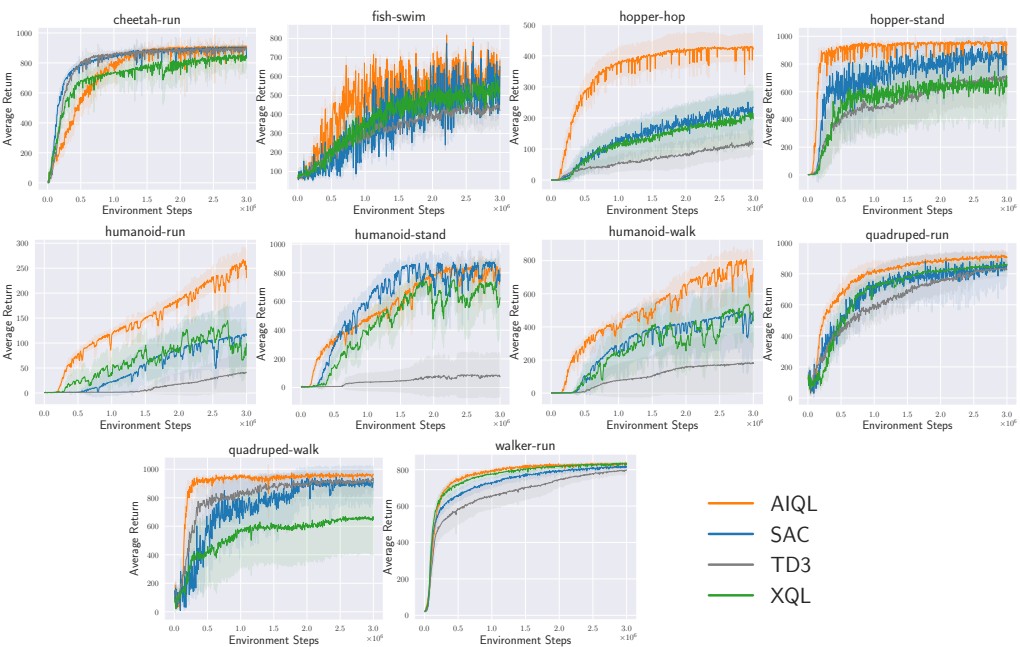

Figure 4: The average return for each task.

including its base algorithm SAC, highlighting the effectiveness of expectile loss and $\tau$ annealing in improving online RL performance.

The scores for each individual task are shown in Figure 4. In tasks like hopper-hop, humanoid-run, and humanoid-walk, where existing methods showed low scores at 3 million steps, AIQL achieved substantial improvements. In tasks such as hopper-stand, quadruped-run, and quadruped-walk, where existing methods attained consistently high scores, AIQL not only achieved high scores with fewer steps but also exhibited reduced variance, resulting in higher final scores. This suggests improved sample efficiency from optimal value estimation and enhanced stability due to increased exploration early in the training. AIQL also outperformed XQL, another method for estimating the optimal value, demonstrating that expectile-based squared loss in online RL is more stable and superior to XQL's exponential-based loss.

### 4.3 EFFECTS OF $\tau$ ANNEALING

Table 3 presents the final average scores for all tasks when $\tau$ is annealed compared to when $\tau$ is fixed. AIQL with $\tau$ annealing, starting from 0.9, outperformed the versions with fixed $\tau$ at 0.6, 0.7, 0.8, 0.9, and 0.95, highlighting the effectiveness of annealing. Figure 5 illustrates the results for two of the challenging tasks: humanoid-run and hopper-hop. In hopper-hop, when $\tau$ is fixed, increasing $\tau$ from 0.5 (SAC) to 0.6 and 0.7 improves the score, but the learning process becomes unstable, leading to a decline in scores at $\tau = 0.8$, with $\tau = 0.7$ producing the best result. In contrast, AIQL, which anneals $\tau$ from 0.9, maintained stable learning and achieved higher final scores than the fixed-$\tau$ version, even with the optimal fixed $\tau$ value

Table 3: The average final score across 10 tasks when $\tau$ is annealed compared to when $\tau$ is fixed. Annealing $\tau$ improves the scores and enhances robustness to $\tau$ settings.

|                  | Mean              | IQM               |
| ---------------- | ----------------- | ----------------- |
| Annealed (0.7)   | $720.2 \pm 55.3$  | $821.7 \pm 21.8$  |
| Annealed (0.8)   | $742.1 \pm 48.0$  | $824.0 \pm 24.4$  |
| Annealed (0.9)   | $\mathbf{746.1} \pm 47.6$ | $\mathbf{832.4} \pm 17.2$ |
| Annealed (0.95)  | $736.1 \pm 48.1$  | $815.6 \pm 22.7$  |
| Fixed (0.6)      | $713.6 \pm 56.2$  | $822.7 \pm 22.1$  |
| Fixed (0.7)      | $730.7 \pm 52.4$  | $825.0 \pm 22.5$  |
| Fixed (0.8)      | $683.4 \pm 56.4$  | $775.9 \pm 27.1$  |
| Fixed (0.9)      | $588.2 \pm 59.2$  | $632.7 \pm 48.6$  |
| Fixed (0.95)     | $364.9 \pm 57.0$  | $303.5 \pm 36.8$  |

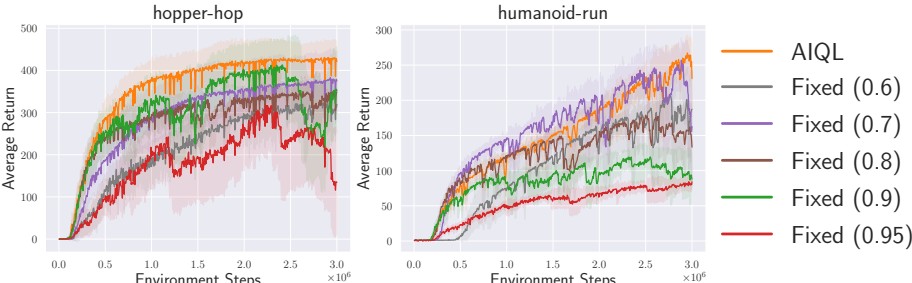

Figure 5: The average return on hopper-hop and humanoid-run when $\tau$ is annealed (AIQL) compared to when $\tau$ is fixed. Annealing $\tau$ improves asymptotic performance and stability.

of 0.7. In humanoid-run, the fixed $\tau = 0.7$ showed better early-stage learning than AIQL, but the learning became unstable over time, resulting in a significant score drop by the end of the training. For larger $\tau$ values, the scores were considerably lower, suggesting high sensitivity to $\tau$ when it is not annealed. AIQL, on the other hand, maintained relatively stable score improvement. The results for other tasks are provided in the appendix.

Even when annealing $\tau$, the initial value of $\tau$ remains a hyperparameter. However, compared to fixed $\tau$, annealing makes the method less sensitive to this hyperparameter. As shown in Table 3, in the fixed $\tau$ case, $\tau = 0.7$ performed best, but increasing it to 0.8 led to a significant drop in scores, with even lower scores for $\tau = 0.9$ and 0.95. In the annealed $\tau$ case, the best initial value was $\tau_{\text{init}} = 0.9$, but scores did not change significantly even when the initial value varied between 0.7 and 0.95, demonstrating that AIQL is more robust to hyperparameter variations compared to the fixed $\tau$ case.

## 4.4 COMPARISON WITH MAX-BACKUP

In continuous action tasks, one straightforward way to compute maximum Q-values is by sampling multiple actions from the current policy, calculating their Q-values, and selecting the maximum. This method, known as max-backup, was employed by Kumar et al. (2020). In this study, we compared max-backup with AIQL using different sampling counts. Figure 6 shows the average scores for humanoid-run and hopper-hop, where AIQL consistently outperformed max-backup across both tasks. This suggests that expectile-based methods for maximum value estimation are more effective than the sampling-based approach used in max-backup. Furthermore, since the computation cost of max-backup increases with the number of samples, AIQL is also superior in terms of both performance and computational efficiency. Results for other tasks are included in the appendix.

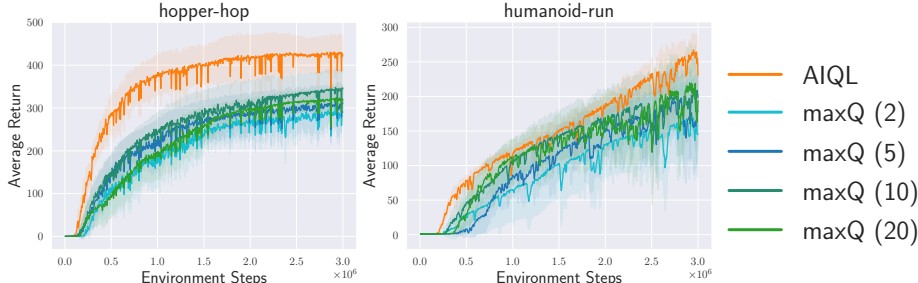

Figure 6: Comparison of AIQL and max-backup in the hopper-hop and humanoid-run tasks. The numbers in parentheses indicate the number of actions sampled to compute the maximum value.

## 4.5 SKEWNESS AND BIAS

Figure 7 (left) shows the skewness for both AIQL and the fixed-$\tau$ cases. When $\tau$ is fixed, skewness increases as learning progresses, with larger $\tau$ values leading to higher skewness, approaching the

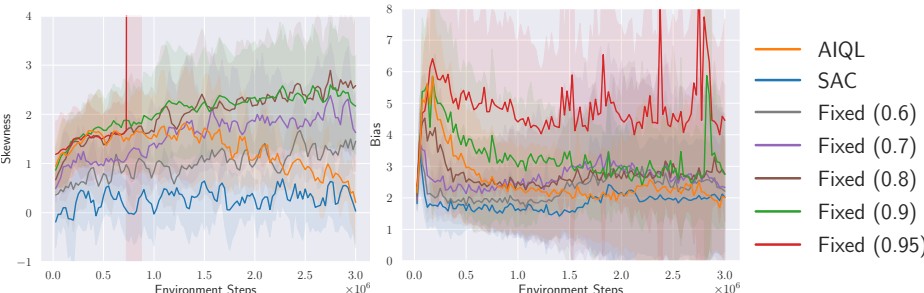

Figure 7: The average bias and skewness across 10 tasks during the learning process. SAC is equivalent to the case where $\tau$ is fixed at 0.5. As $\tau$ increases, skewness gradually becomes larger, and the final bias also increases. In AIQL, skewness decreases toward the end of learning, which helps suppress the increase in bias. In the figure on the left, the plot is terminated at the point where the values diverge.

operation of selecting the maximum. This is similar to extreme value theory, causing more significant distortion in the Q-value distribution. In AIQL, while skewness gradually increases early in learning, annealing reduces skewness over time, preventing the consistent increase seen in the fixed-$\tau$ case.

The bias, also depicted in Figure 7 (right), shows that in the fixed-$\tau$ case, bias increases significantly in the early stages of learning. For $\tau$ values below 0.9, it temporarily decreases before gradually rising again as learning progresses. For $\tau = 0.9$ and 0.95, bias remains high after the initial rise and spikes sharply toward the end of training. As discussed in Section 3.2, an increase in skewness may be one of the potential causes of overestimation bias. Even in $\tau = 0.5$ (SAC), bias increases, likely due to policy improvement updates as explained in Fujimoto et al. (2018). In AIQL, although bias increases initially, annealing prevents it from rising further during training, keeping it relatively constant and allowing for stable learning.

## 4.6 OTHER ANNEALING PATTERNS

We also tested non-linear annealing patterns. Figure 8 illustrates different annealing patterns, including exponential annealing as proposed by Morerio et al. (2017) (Exp1), its inverse (Exp2), and sigmoid-based annealing (Sigmoid). The results, as shown in Table 4, indicate that Sigmoid and Exp1 performed similarly to linear annealing, but Exp2 resulted in worse performance. This suggests that prolonging the high $\tau$ period leads to excessive initial bias, which negatively impacts the later stages of learning. These observations indicate that linear annealing proves to be effective enough, and future research could explore dynamic adjustments based on bias or skewness.

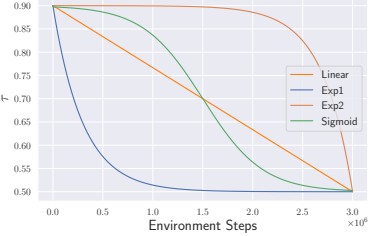

Figure 8: The annealing patterns of $\tau$ used in the experiments.

Table 4: The average score across 10 tasks for various annealing patterns. A simple linear annealing showed the best performance.

|  | Mean | IQM |
|---|---|---|
| Linear | $746.1 \pm 47.6$ | $832.4 \pm 17.2$ |
| Exp1 | $728.1 \pm 50.0$ | $809.1 \pm 26.9$ |
| Exp2 | $694.9 \pm 53.2$ | $772.4 \pm 32.8$ |
| Sigmoid | $742.2 \pm 46.6$ | $812.9 \pm 26.4$ |

## 5 RELATED WORK

**Off-Policy Online RL with Continuous Action Space**   In continuous action space online RL tasks, methods such as SAC (Haarnoja et al., 2018) and TD3 (Fujimoto et al., 2018) are widely used. SAC is an actor-critic method based on Haarnoja et al. (2017) that incorporates entropy maximization. The value function adds the policy's entropy to the target value and models the soft value function for the current policy. In policy learning, SAC aims for policy improvement by aligning the policy distribution with the exponential of the Q-function. TD3 is an extension of DDPG (Lillicrap, 2015) that incorporates techniques like clipped double Q-learning, action noise, and delayed policy updates. Like SAC, TD3 updates the value function based on SARSA, learning the value function under the current policy. The policy itself is improved using the deterministic policy gradient, as in DDPG. Both methods employ SARSA-based updates, focusing on learning the value function for the current policy. Our study demonstrates that by replacing SAC's value function update with expectile loss, shifting to a Q-learning-based maximization, we can achieve improved performance.

Overestimation bias is a recurring issue in RL. Thrun & Schwartz (1993) formalized how noise in function approximators leads to overestimation bias, showing how noise variance and the number of actions contribute to the increase in bias. Techniques like Lan et al. (2020) and Chen et al. (2021) acknowledge that overestimation can sometimes be beneficial for exploration, while using ensembles to control the mean and variance of bias. Incorporating these ensemble techniques into our proposed method could potentially enhance sample efficiency, which could be a promising direction for future work.

**Value Function Maximization in Continuous Action Space**   In continuous action tasks, calculating the maximum Q-value is generally intractable. Various methods have tackled this issue by using asymmetric loss functions (Kostrikov et al., 2022; Garg et al., 2023; Xu et al., 2023; Omura et al., 2024; Sikchi et al., 2024), evaluating multiple actions (Kalashnikov et al., 2018; Kumar et al., 2019; 2020), or discretizing actions (Tavakoli et al., 2018; Seyde et al., 2023; 2024).

IQL (Kostrikov et al., 2022) approaches Q-value as a random variable with inherent randomness related to the actions, modeling the maximum value using expectile loss with an expectile parameter close to 1. By adjusting the hyperparameter $\tau$ from 0.5 to 1, the value function estimate shifts from SARSA-based to Q-learning-based. XQL (Garg et al., 2023) models the soft optimal value in maximum entropy RL using Gumbel loss, derived from maximum likelihood estimation under the assumption of a Gumbel error distribution. MXQL (Omura et al., 2024) stabilizes the Gumbel loss by employing a Maclaurin expansion. By increasing the order of the expansion from 2 to infinity, MXQL transitions from SARSA-based learning to (soft) Q-learning-based learning. Although MXQL may benefit from annealing the expansion order to improve performance, tuning hyperparameters can be challenging due to the infinite range of the optimality parameter.

Kalashnikov et al. (2018); Kumar et al. (2019; 2020) improve sample efficiency by modeling the optimal Q-value through sampling multiple actions from the policy distribution and selecting the action that maximizes the Q-value. These methods can shift from SARSA-based to Q-learning-based by increasing the number of samples from 1 to infinity, but tuning poses challenges, and computational costs rise significantly as the number of samples increases.

Some methods, such as (Tavakoli et al., 2018; Seyde et al., 2023; 2024), achieve optimal Q-value estimation through action discretization. However, if the number of intervals is small, accurate action selection becomes difficult, while increasing the intervals reduces sample efficiency.

In this study, we chose expectile loss for optimal value estimation and annealing due to its relatively simple hyperparameter tuning and consistent computational complexity compared to these other approaches.

## 6 CONCLUSION

In this study, we evaluated the effectiveness of a method that uses expectile loss to model the optimal Q-value in online RL for continuous action tasks. We demonstrated that the distortion of the Q-value distribution and overestimation bias can be amplified based on the level of optimality. Based on this, we proposed an annealing method for adjusting optimality, which promotes exploration in

the early stages of learning and maintains stability in the later stages. The proposed Annealed Implicit Q-learning (AIQL) outperformed existing algorithms such as SAC and TD3. Additionally, the annealing process stabilized learning and improved robustness to hyperparameter changes. While a simple linear annealing method was found to contribute to performance improvement, further optimization of methods to adjust optimality is a promising direction for future research.

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

## A EXPERIMENTAL DETAILS

### A.1 PRELIMINARY EXPERIMENTS

By using the inverse Gaussian distribution, various distributions with constant variance and mean but different skewness were realized. The PDF of the inverse Gaussian distribution is as follows.

$$f(x; \mu, \lambda) = \left(\frac{\lambda}{2\pi x^3}\right)^{\frac{1}{2}} \exp\left(-\frac{\lambda(x-\mu)^2}{2\mu^2 x}\right).$$

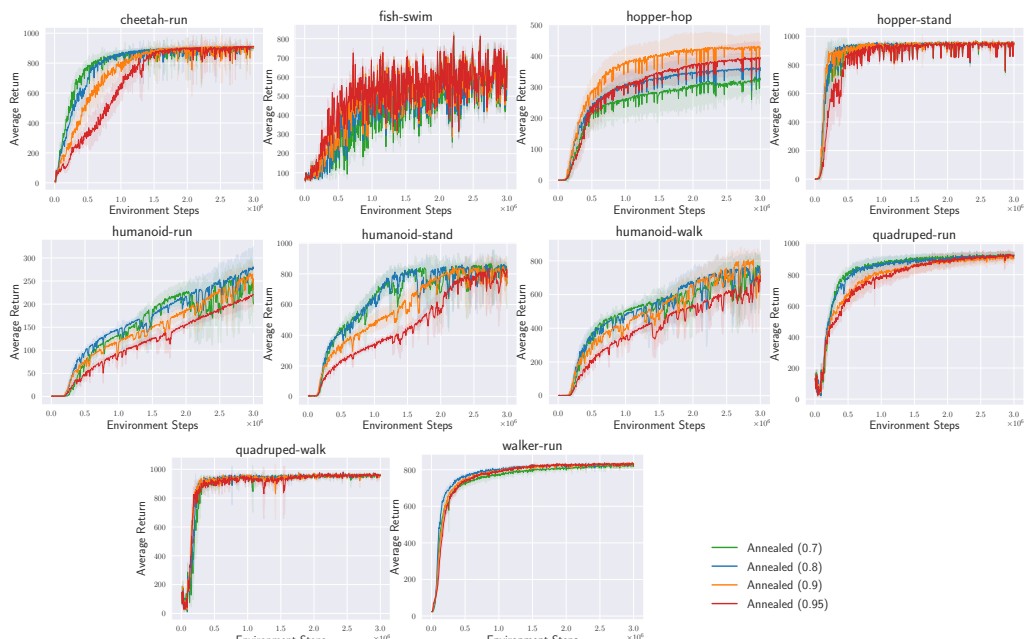

Figure 9: The average return for each task when annealing with various $\tau_{\text{init}}$ values.

The mean is $\mu$, the variance is $\sigma^2$, and the skewness $\gamma$ is expressed as $\sigma^2 = \frac{\mu^3}{\lambda}$ and $\gamma = \frac{3\sigma}{\mu}$, respectively. Therefore, $\mu$ and $\lambda$ can be expressed using $\sigma$ and $\gamma$ as follows:

$$\mu = \frac{3\sigma}{\gamma},$$

$$\lambda = \frac{\mu^3}{\sigma^2} = \frac{(3\sigma/\gamma)^3}{\sigma^2} = \frac{27\sigma}{\gamma^3}.$$

Thus, by setting $\sigma = 1$ and using a given $\gamma$, $\mu$ and $\lambda$ are calculated. By sampling from appendix A.1 and subtracting $\mu$ from those values, it is possible to obtain samples from a distribution with mean 0, standard deviation 1, and arbitrary skewness. For each skewness, 100 samples are drawn, and the maximum value is calculated. This process is repeated 3,000 times, and the mean and standard deviation of these 3,000 maximum values are plotted in Figure 2.

## A.2 Experiments on DM Control

AIQL is built upon SAC, and the SAC implementation follows D'Oro et al. (2023). To ensure a fair comparison, all methods employed a batch size of 256, and both the actor and critic networks used two hidden layers consisting of 256 units each. TD3 and XQL use their respective official implementations. For XQL, we used XSAC, which integrates XQL with SAC. As described in the XQL paper, the temperature parameter $\beta$ was evaluated for values [1, 2, 5], and the best value $\beta = 5$ was chosen. Other hyperparameters for all methods follow the values reported in their respective papers. For all experiments, the results are averaged over 10 random seeds.

## B Experimental Results Using Fixed $\tau$ and Annealed $\tau$

Figure 9 shows the average return for each task when annealing from various $\tau_{\text{init}}$ values. Figure 10 shows the average return for each task when using various fixed $\tau$ values. When annealing, the performance is less sensitive to the $\tau$ value, demonstrating increased robustness to hyperparameters through annealing.

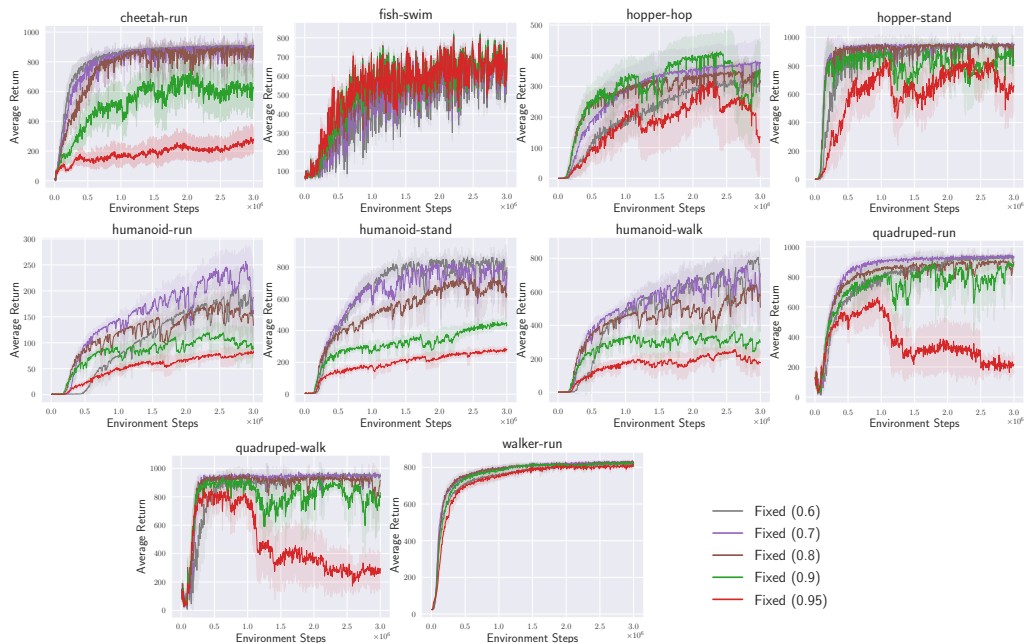

Figure 10: The average return for each task when using various fixed $\tau$ values.

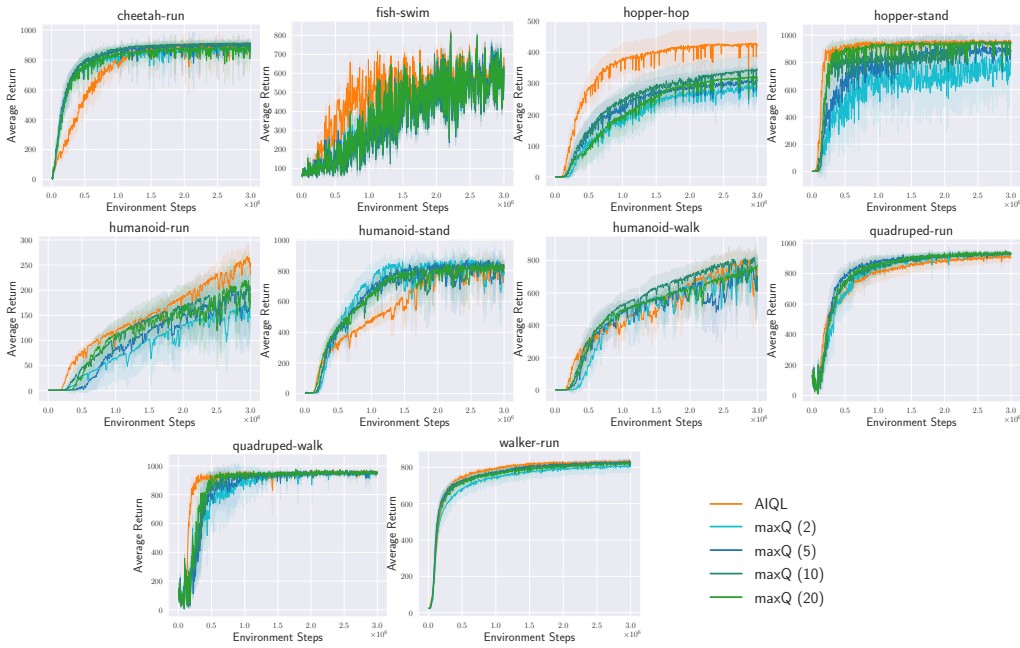

Figure 11: The average return for each task when using max-backup with various action sampling numbers.

## C   EXPERIMENTAL RESULTS OF MAX-BACKUP

Figure 11 presents the average return for each task when using max-backup with various action sampling numbers. Max-backup is implemented based on Kumar et al. (2020). While max-backup has the drawback of increased computational cost, it demonstrated scores comparable to AIQL in some tasks. However, in more challenging tasks such as hopper-hop and humanoid-run, AIQL outperformed max-backup.

## D INTEGRATION OF AIQL AND TD3

To demonstrate the versatility of AIQL, experiments were conducted combining it not only with SAC but also with TD3. The results are shown in Figure 12. Similar to the case with SAC, performance is significantly improved when combined with TD3.

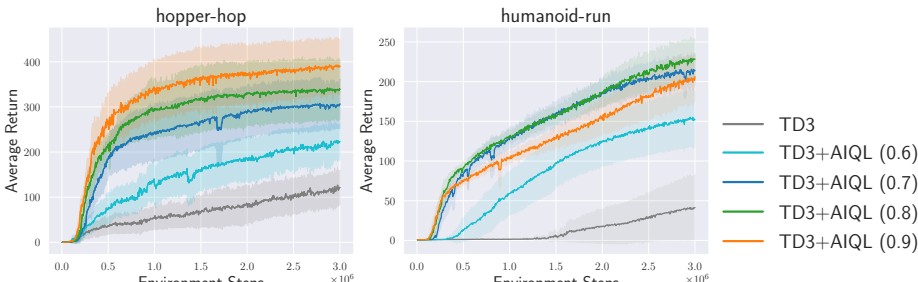

Figure 12: The average return of the method combining AIQL with TD3. The values in parentheses represent $\tau_{\text{init}}$.

## E ANALYSIS OF ANNEALING DURATION

Figure 13 illustrates the results of varying the annealing duration $T$ in AIQL combined with SAC. After the annealing phase, learning proceeds with $\tau = 0.5$, consistent with the standard SAC setting. Notably, even when $T$ is as small as 1 million steps, there is a significant performance improvement compared to SAC. The final return does not vary significantly across different values of $T$, indicating that AIQL is robust to $T$ within the range tested.

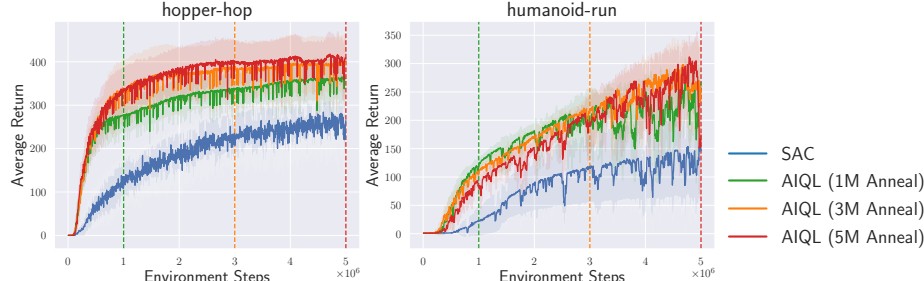

Figure 13: The average return of AIQL when the annealing duration is varied. The dashed line represents the step count at which annealing ends, after which learning proceeds with $\tau = 0.5$, the same as the SAC.

## F THE NECESSITY OF THE $V$-FUNCTION

In IQL, both the Q-function and V-function were trained to handle environmental stochasticity. However, as shown in Figure 14, this study confirms that training with only the Q-function achieves better performance even in stochastic environments. Therefore, AIQL utilizes only the Q-function.

Figure 14 presents results comparing AIQL using only the Q-function versus AIQL using both the Q-function and V-function in the stochastic hopper-hop environment. Stochasticity in the environment is introduced by adding Gaussian noise with a mean of 0 to the actions input into the DM Control tasks. The left figure shows the results when $\tau$ is annealed, while the right figure corresponds to fixed $\tau$. The values used were $\tau_{\text{init}} = 0.9$ for annealing and $\tau = 0.7$ for the fixed case, as these settings yielded the best performance.

In both cases, whether annealing or fixed, using only the Q-function consistently outperformed using both the Q-function and V-function across all levels of noise standard deviation. This performance difference is attributed to the additional network required when using the V-function, which increase approximation error and degrade performance.

For the annealing case in the left figure, the degradation in average return as noise standard deviation increases is similar whether or not the V-function is used. This is likely because annealing results in learning behavior similar to SAC toward the end of training, mitigating the impact of environmental randomness on the IQL loss.

In the fixed $\tau$ case shown in the right figure, performance degradation due to increased noise standard deviation is more pronounced when the V-function is not used. Nevertheless, using only the Q-function still outperforms the setup with both Q-function and V-function.

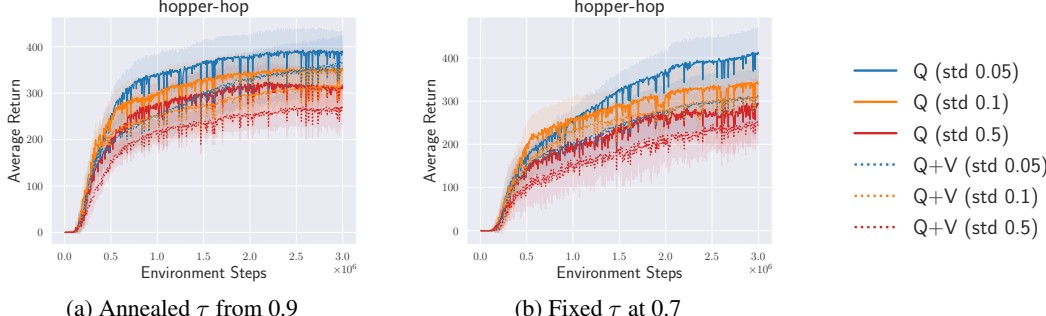

(a) Annealed $\tau$ from 0.9          (b) Fixed $\tau$ at 0.7

Figure 14: The average scores for AIQL trained using only the Q-function compared to AIQL utilizing both the Q-function and V-function in a stochastic hopper-hop environment. The stochastic environment is created by adding zero-mean Gaussian noise to the actions fed into the DM Control environment. The different colors represent the varying standard deviations of the Gaussian noise applied. The solid lines represent results obtained using only the Q-function, while the dotted lines indicate those obtained using both the Q-function and V-function. The left figure shows the results when $\tau$ is annealed from an initial value of 0.9, and the right figure shows the results when $\tau$ is fixed at 0.7.

