# OpenReview forum: "Annealed Implicit Q-learning in Online Reinforcement Learning"
_ICLR.cc/2025/Conference — Submitted to ICLR 2025_

### Official Review · Reviewer_4Sx9 · 2024-10-17

**Soundness:** 3
**Presentation:** 3
**Contribution:** 2
**Rating:** 6
**Confidence:** 4

**Summary:**

The authors propose a sample-efficient actor-critic method by modifying the policy evaluation loss in online scenarios. They propose to do so by leveraging the expectile loss shown effective in offline scenarios. Additionally, the authors propose to decrease the learned expectile during training to perform implicit Q-learning at the beginning of the training and transition to a policy evaluation regime at the end of the training.

**Strengths:**

1. The paper investigates an important topic in RL.

2. The proposed method is clearly explained and well-motivated.

3. The presented algorithm outperforms relevant baselines in the considered experiments.

**Weaknesses:**

1. The novelty of the proposed approach is limited as the paper's main contribution lies in applying Implicite Q-Learning to an online setting with a linear schedule for the expectile. Nevertheless, the findings are interesting. They would be worth sharing with the community if a more detailed study would be done. Here are some ideas to increase the depth of the analysis:

     ~a.  The authors should comment on the choice of learning the Q-function directly instead of using a separate value function as done in IQL. This choice should normally harm the performances as the learned expectile is now influenced by the stochasticity of the environment. Evaluating the proposed algorithm in a highly stochastic MDP would help justify this choice.~

     ~b. AIQL is only applied to SAC. Applying it to TD3 and CrossQ, for example, would strengthen the authors' claim.~

     c. Investigating the importance of the expectile regression loss compared to the quantile or Huber quantile regression loss would be interesting.

     ~d. Analyzing the influence of the hyperparameter $T$ would be beneficial for understanding its influence on AIQL.~

2. Changing the target expectile value during the training makes the loss non-stationary, potentially harming performances. The authors do not comment on this point.

~3. The code is not provided, which limits the reliability of the experiments. Additionally, the baselines’ hyperparameters are not reported.~

**Questions:**

~1. Line 155, the authors claim that the scenario where all action values are equal has the largest overestimation bias. Can this be justified?~

2. To mitigate the issue presented in Weakness 2, the authors could consider having a Q-network with several heads, where each head is responsible for learning one expectile, similar to Quantile-Regression Deep Q-Network [1]. The head used for training the policy would evolve during training, starting from a low target expectile value to finish with a high target expectile value as proposed by the authors. I would be interested in the authors' comments about this suggestion.

[1] Dabney, Will, et al. "Distributional reinforcement learning with quantile regression." AAAI. 2018.

– Remarks –

~A. Line 35, a stop between “efficiency” and “This” is missing.~

~B. Line 42, for clarity, I would replace “optimal value” by “maximal value”.~

~C. Line 90, the word “initial” is missing before “probability distribution”.~

~D. Line 120, the value function should not take the action $a$ as input.~

~E. Line 196, some parentheses are close, but they have never been opened. I suggest removing them.~

~F. Line 215, replacing $t$ by $\min(t, T)$ in the right side of the equation would be more accurate.~

---

> ### Author Response · Authors · 2024-11-21
> **Response (1/1)**
>
> Thank you very much for your valuable feedback and comments! Below, we provide our responses.
>
> **Q1. The authors should comment on the choice of learning the Q-function directly instead of using a separate value function as done in IQL. This choice should normally harm the performances as the learned expectile is now influenced by the stochasticity of the environment. Evaluating the proposed algorithm in a highly stochastic MDP would help justify this choice.**
>
> We conducted experiments in stochastic environments to compare the performance of AIQL when using both V-function and Q-function versus using only Q-function, and we have added the results in Section F. The stochastic environments were created by adding Gaussian noise to the actions input into the DM Control tasks. Even when significant noise (standard deviation of 0.5) was added to the action with
>  range [-1, 1], the Q-function-only AIQL achieved better performance. Additionally, the annealing mechanism effectively mitigated the performance degradation caused by the noise. These results support the validity of our implementation, which excludes the V-function. We will include experiments on other tasks in the camera-ready version.
>
> **Q2. AIQL is only applied to SAC. Applying it to TD3 and CrossQ, for example, would strengthen the authors' claim.**
>
> CrossQ is known to be sensitive to hyperparameters and environment settings [2], making it less generalizable. Therefore, we conducted additional experiments with TD3 and added the results in Section D. By simply replacing the critic's L2 loss in TD3's official implementation, we observed similar performance improvements to those seen with SAC in the hopper-hop and humanoid-run tasks. We will include results for other tasks.
>
> **Q3. Analyzing the influence of the hyperparameter T would be beneficial for understanding its influence on AIQL.**
>
> We have added results in Section E showing the effects of varying the annealing duration $T$. Interestingly, even with a small $T$ of 1M steps, AIQL still achieved significant performance improvements compared to SAC. Within the range of $T$ tested in these experiments, AIQL demonstrated a certain degree of robustness to changes in $T$.
>
> **Q4. The code is not provided, which limits the reliability of the experiments. Additionally, the baselines’ hyperparameters are not reported.**
>
> We have uploaded the code. Additionally, we have updated Section A.2 with details about the baseline hyperparameters.
>
> **Q5 Line 155, the authors claim that the scenario where all action values are equal has the largest overestimation bias. Can this be justified?**
>
> Here is an intuitive example. Consider two scalar values $a$ and $b$ as the true Q-values for two actions, where the noise $e_1$ and $e_2$ is added to each value. The $e_1$ and $e_2$ follow a uniform distribution $U(−1,1)$. In this case, the overestimation bias can be expressed as:
>
> $Z = \mathbb{E}_{e_1,e_2}[\max(a+e_1,b+e_2)]-\max(a,b).$
>
> When $a \geq b$, overestimation does not occur if only $a + e_1$ is chosen in $\max(a + e_1, b + e_2)$ because $Z = \mathbb{E}[a+e_1]-a=0$. However, due to the influence of noise $e_1$ and $e_2$, there are instances where $b + e_2$ can be larger than $a + e_1$, resulting in overestimation.
>
> This means that when $a$ and $b$ are far apart, $\max(a + e_1, b + e_2)$ is more likely to be $a + e_1$. On the other hand, when $a$ and $b$ are close, $\max(a + e_1, b + e_2)$ is more likely to select  $b + e_2$, increasing the chances of overestimation. Overestimation reaches its maximum when $a = b$.
>
> For example, if $a = 2$ and $b = 0$, $Z = 2 - 2 = 0$, meaning no overestimation occurs. However, if $a = b = 2$, then:
>
>
> $ Z=\mathbb{E}_{e_1,e_2}[\max (2+e_1,2+e_2)]-2 $
>
> $ = \mathbb{E}_{e_1,e_2}[\max (e_1, e_2)]. $
>
>
> In this case, overestimation occurs. This type of formulation was derived in [3], which may serve as a useful reference.
>
> **Q6. To mitigate the issue presented in Weakness 2, the authors could consider having a Q-network with several heads, where each head is responsible for learning one expectile, similar to Quantile-Regression Deep Q-Network [1]. The head used for training the policy would evolve during training, starting from a low target expectile value to finish with a high target expectile value as proposed by the authors. I would be interested in the authors' comments about this suggestion.**
>
> Thank you very much for the highly intriguing suggestion! While it may lead to an increase in computational cost, we agree that using a distributional critic to achieve a stationary loss is an effective approach. (We avoid discrete changes in the loss by not using max-backup [4] or MXQL [5], instead opting for expectile loss to ensure continuous changes.) Although our computational resources are limited and results are not yet available, we plan to include the results in the camera-ready version.
>
> **Q7. Remarks**
>
> Thank you for pointing out those mistakes! We have corrected them.

---

> ### Author Response · Authors · 2024-11-21
> **References**
>
> [2] Michal Nauman, , Mateusz Ostaszewski, Krzysztof Jankowski, Piotr Mi\los, Marek Cygan. Bigger, Regularized, Optimistic: scaling for compute and sample efficient continuous control. The Thirty-eighth Annual Conference on Neural Information Processing Systems. 2024.
>
> [3] Sebastian Thrun and Anton Schwartz. Issues in using function approximation for reinforcement learning. In Fourth Connectionist Models Summer School, 1993.
>
> [4] Aviral Kumar, Aurick Zhou, George Tucker, and Sergey Levine. Conservative q-learning for offline reinforcement learning. In H. Larochelle, M. Ranzato, R. Hadsell, M.F. Balcan, and H. Lin (eds.), Advances in Neural Information Processing Systems, volume 33, pp. 1179–1191. Curran Associates, Inc., 2020.
>
> [5] Motoki Omura, Takayuki Osa, Yusuke Mukuta, and Tatsuya Harada. Stabilizing extreme q-learning by maclaurin expansion. In Reinforcement Learning Conference, 2024.

---

> > ### Comment · Reviewer_4Sx9 · 2024-11-22
> >
> > I thank the Authors for their detailed answers. I have increased the score as most of my concerns have been addressed. Aside from the remaining concerns, I further refine the following point:
> >
> > > Q5 Line 155, the authors claim that the scenario where all action values are equal has the largest overestimation bias. Can this be justified?
> >
> > The authors' justification assumes that the noise added to the "true" value is i.i.d. However, in the submission, this assumption is only made after claiming that the scenario where all action values are equal has the largest overestimation bias. This assumption needs to be stated before this affirmation, as there exist some other noise models with which this affirmation is wrong.
> >
> > Additionally, including pointers from the main text to the newly added material in the appendix would strengthen the paper.

---

> > > ### Author Response · Authors · 2024-11-28
> > > **Additional Response**
> > >
> > > Thank you for your valuable feedback. We have added a description of the i.i.d. nature of the noise and included pointers to the additional experiments. if you have any further concerns, please do not hesitate to let us know.

---

### Official Review · Reviewer_N256 · 2024-10-31

**Soundness:** 4
**Presentation:** 3
**Contribution:** 3
**Rating:** 6
**Confidence:** 3

**Summary:**

This paper proposes a new approach to improve sample efficiency in continuous action online RL by gradually reducing optimality bias, which helps control overestimation in Q-values. The method outperforms SAC and TD3 on DM Control tasks, offering better performance and robustness.

**Strengths:**

- Tackles a key challenge in online RL: boosting sample efficiency through better value function estimation.
- Presents a clear argument that overestimation and Q-value skewness are major issues, and introduces expectile loss in the critic architecture to help address them in continuous action tasks.
- Shows impressive performance gains compared to popular methods like SAC and TD3.

**Weaknesses:**

- Section 2.2 (Line 115): Why is it stated that "In IQL, near-optimal values are estimated by using τ close to 1"? Could you clarify this choice?
~~- Sections 4.1 & 4.6: Could you expand your analysis to discuss Exp1 in Section 4.6? Based on the explanation in Section 4.1, it seemed the natural annealing strategy might align with Exp1 in Section 4.6, yet it performed relatively worse than the linear and sigmoid strategies.~~
~~- Section 4.2: In the XQL (Garg et al., 2023), the extension of SAC (X-SAC) showed comparable performance to SAC, even outperforming it in Hopper-Hop. However, the results in Figure 4 don’t seem to align with this finding. Could you provide any reasoning or justification for this discrepancy?~~

**Questions:**

none

---

> ### Author Response · Authors · 2024-11-21
> **Response (1/1)**
>
> Thank you for your valuable comments! Here are our responses to them.
>
> **Q1. Section 2.2 (Line 115): Why is it stated that "In IQL, near-optimal values are estimated by using τ close to 1"? Could you clarify this choice?**
>
> The goal of IQL is to compute the optimal target value ($r + \max Q(s',a')$) for continuous action tasks. In IQL's expectile loss, setting $\tau = 1$ allows for handling this optimal target value directly. However, due to issues such as overestimation bias and stability concerns, values like $\tau = 0.7$ or $0.9$ are typically used to estimate a near-optimal value instead.
>
> **Q2. Sections 4.1 & 4.6: Could you expand your analysis to discuss Exp1 in Section 4.6? Based on the explanation in Section 4.1, it seemed the natural annealing strategy might align with Exp1 in Section 4.6, yet it performed relatively worse than the linear and sigmoid strategies.**
>
> An annealing method like Exp1, which reduces $\tau$ more significantly at the beginning, might potentially improve performance further. However, Exp1's initial decay is too aggressive, suggesting that an annealing strategy falling between Exp1 and Linear could be more effective. Despite this, we demonstrated that a straightforward linear annealing approach achieves strong performance without requiring detailed tuning of the decay method. Discovering the optimal annealing strategy remains a promising direction for future research.
>
> **Q3. Section 4.2: In the XQL (Garg et al., 2023), the extension of SAC (X-SAC) showed comparable performance to SAC, even outperforming it in Hopper-Hop. However, the results in Figure 4 don’t seem to align with this finding. Could you provide any reasoning or justification for this discrepancy?**
>
> In our study, to ensure a fair comparison, we set the batch size and the number of hidden units in the network to 256, following the configuration used in the SAC paper. In the XQL paper, as noted in Appendix D, the batch size and the number of hidden units were set to 1024, which may have influenced the results. For XQL in our study, we used the official implementation of XQL and kept all other hyperparameters consistent with those reported in the paper. We have added details about the baseline hyperparameters in Section A.2.

---

> ### Author Response · Authors · 2024-11-28
>
> We have included new experiments in Appendix D, E, and F to provide stronger support for the claims in our paper. In addition, we have expanded the explanations in the Introduction and Section 3.3 to address any potential ambiguity regarding the motivation and claims. We would be grateful if you could review these updates and our earlier response and let us know your thoughts.

---

> > ### Comment · Reviewer_N256 · 2024-12-01
> > **Thanks for the additional details.**
> >
> > Thank you for your clarification. However, I am still unclear on why issues such as overestimation bias and stability concerns specifically lead to the choice of $\tau = 0.7$ or $0.9$. Could you elaborate on how these values mitigate those concerns compared to $\tau = 1$?
> >
> > The other two questions have been satisfactorily addressed by your response—thank you!

---

> > > ### Author Response · Authors · 2024-12-02
> > >
> > > Thank you for your question! In the IQL loss, when τ equals 1, it corresponds to a Q-learning-based target using the max operator $r + \gamma \max Q(s', a')$ , whereas when τ equals 0.5, it corresponds to a SARSA-based on-policy target $ r + \gamma \mathbb{E}_{a' \sim \pi} Q(s', a')$ . By varying τ from 0.5 to 1 in the IQL loss, we can smoothly transition from a SARSA-based loss to a Q-learning-based loss. Since overestimation arises from the max operator, it becomes more likely to occur as τ approaches 1. We have added this explanation to Section 3.3. Furthermore, as shown in Figure 1 of our paper, the shape of the loss function becomes more asymmetric as τ increases. When τ equals 1, as can be seen from the loss equation, the penalty for target values smaller than the estimated value becomes zero, leading to progressively larger estimated values. To avoid estimating excessively large values in this way, values such as 0.7 and 0.9 are used in IQL.
> > >
> > > Please let us know if there are any other points that require explanation!

---

### Official Review · Reviewer_9sVB · 2024-10-31

**Soundness:** 2
**Presentation:** 2
**Contribution:** 2
**Rating:** 3
**Confidence:** 4

**Summary:**

The paper proposes a new method in online RL, focusing on the setting with continuous action space. The work starts from the advantage of expectile and the observation of overestimation bias in function approximation. Then a new algorithm named Annealed Implicit Q-learning (AIQL) is proposed, which is based on Soft Actor-Critic (SAC) and the tool of expectile inspired by Implicit Q-learning (IQL). Experiments are then to show the comparisons in performances.

**Strengths:**

> **Clarity**
- Despite some lack of notation explanation, the paper gradually leads readers to the proposed method through observation of gaps and theoretical discussion, making the main method easy to understand.

> **Significance**
- The proposed method is tested in various benchmark environments.

**Weaknesses:**

> **Originality**
- The proposed method mainly follows the ideas of both IQL and SAC.

> **Quality**
- The claims regarding the overestimation bias are not sufficiently convincing. Equation (1) is derived following the strong assumption that Q-functions are independent from actions, which tends to be an over-simplification of the scenario. In addition, the experiment shown in Figure 2 mainly concerns a single training step, but does not explain why such bias can still accumulate after online exploration (apart from claiming such result by 'if the bias becomes too large').
- In Section 3.3, the advantage of tuning $\tau$ is described, after which a linearly decreasing $\tau(t)$ is proposed. However, there is neither theoretical/empirical observation showcasing how different $\tau$ affects the training in different stages (as a support to the advantage part), nor justification of the choice of linear functions (as a support to the proposal part).

> **Clarity**
- In Section 2.1, $d\_0$ is not clearly defined by just claiming it is 'the probability distribution', which actually seems to be the initial state distribution. And the horizon $T$ is not defined.
- In the review of IQL, the loss function for the parameters of Q-functions misses the dependence on $V\_\psi$.

> **Significance**
- In Section 3.1 the paper emphasizes its advantage by estimating directly the optimal value compared to methods like SAC, while in actual implementation according to Section 4.1, the proposed method is just an expectile-loss version of SAC. More clarifications on distinctions between methods could be helpful.
- In some experimental results in Figure 4, SAC reaches higher average returns earlier than AIQL.

**Questions:**

- Why is there no comparison with some online version of IQL, if any?

---

> ### Author Response · Authors · 2024-11-21
> **Response (1/2)**
>
> Thank you for your insightful comments. Below, we provide our responses to your remarks.
>
> **Q1. The proposed method mainly follows the ideas of both IQL and SAC.**
>
> In our study, we found that the simple application of IQL loss to online RL significantly improves performance. While the approach is straightforward, it represents a novel finding that has not yet been explored, which we believe is a major contribution. To demonstrate the generality of this approach, we have conducted additional experiments combining it not only with SAC but also with TD3 and added the results in Section D. Similar performance improvements were observed in the case of TD3 as well.
>
> **Q2. The claims regarding the overestimation bias are not sufficiently convincing. Equation (1) is derived following the strong assumption that Q-functions are independent from actions, which tends to be an over-simplification of the scenario. In addition, the experiment shown in Figure 2 mainly concerns a single training step, but does not explain why such bias can still accumulate after online exploration (apart from claiming such result by 'if the bias becomes too large').**
>
> The simplified formulation is designed to analytically derive the overestimation bias, and its simplicity does not imply a lack of relevance to actual learning. The same formulation has been employed in works such as [1, 2, 3], and in [2, 3], the bias adjustments derived from this formulation have been shown to be effective in actual learning scenarios. While it is true that bias can be somewhat mitigated during online exploration, as shown in Figure 7 of our study, skewness accumulates progressively during learning, and bias, after an initial decrease in the early stages of learning, gradually increases over time
>
> **Q3. In Section 3.3, the advantage of tuning τ is described, after which a linearly decreasing τ(t) is proposed. However, there is neither theoretical/empirical observation showcasing how different τ affects the training in different stages (as a support to the advantage part), nor justification of the choice of linear functions (as a support to the proposal part).**
>
> Figure 7 demonstrates that skewness and bias increase when $\tau$ is fixed, providing the experimental motivation for annealing. While it is possible to consider annealing schedules other than linear one, such as dynamically adjusting $\tau$ based on bias or skewness, we prioritized simplicity in our study. Thus, we adopted linear annealing, which has low computational cost and requires fewer hyperparameters. (In contrast, scheduling based on bias or skewness may introduce additional computation and require extra hyperparameters.)
>
> **Q4. In Section 2.1, d0 is not clearly defined by just claiming it is 'the probability distribution', which actually seems to be the initial state distribution. And the horizon T is not defined. In the review of IQL, the loss function for the parameters of Q-functions misses the dependence on Vψ.**
>
> Thank you for pointing out those mistakes! We have corrected them accordingly.
>
> **Q5. In Section 3.1 the paper emphasizes its advantage by estimating directly the optimal value compared to methods like SAC, while in actual implementation according to Section 4.1, the proposed method is just an expectile-loss version of SAC. More clarifications on distinctions between methods could be helpful.**
>
> Our proposed method is simple: we replace the L2 loss used in the critic learning of SAC and TD3 with expectile loss and introduce annealing for $\tau$. When using L2 loss in critic learning, the target value is based on the SARSA framework ($r + \gamma \mathbb{E}[Q(s',a')]$). By using expectile loss, we instead estimate the near-optimal value ($r + \gamma \max[Q(s',a')]$). This straightforward modification leads to performance improvements and represents a key contribution of our work. Moreover, it is a generalizable approach that can be combined not only with SAC but also with other methods.
>
> **Q6. In some experimental results in Figure 4, SAC reaches higher average returns earlier than AIQL.**
>
> In the cheetah-run and humanoid-stand tasks, SAC shows slightly better performance than AIQL during the early stages of training (e.g., at 1 million steps). However, in the other 8 tasks, AIQL outperforms SAC, as can also be observed from the average performance in Table 1, where AIQL demonstrates a clear advantage. For cheetah-run and humanoid-stand, it is possible that their sensitivity to bias causes the initial bias to slow down learning. Nevertheless, due to the annealing mechanism, AIQL achieves similar final scores to SAC in these tasks.

---

> > ### Comment · Reviewer_9sVB · 2024-11-25
> >
> > Thanks for the point-by-point responses! Please see the follow-up concerns/questions:
> >
> > > **The simplified formulation is designed to analytically derive the overestimation bias, and its simplicity does not imply a lack of relevance to actual learning. The same formulation has been employed in works such as [1, 2, 3], and in [2, 3], the bias adjustments derived from this formulation have been shown to be effective in actual learning scenarios. While it is true that bias can be somewhat mitigated during online exploration, as shown in Figure 7 of our study, skewness accumulates progressively during learning, and bias, after an initial decrease in the early stages of learning, gradually increases over time**
> >
> > - Motivated by such over-simplified formulation, as suggested in the references, solutions could help adjust bias. But that does not mean the theoretical analysis itself has enough capability in modelling the actual settings. Still, it's good to know the current literature with similar approaches.
> >
> > - Seems that the authors agree that no theoretical analysis has been provided for why such biases accumulate in the training.
> >
> > - According to Figure 7., SAC performs the best in most training steps. Does it mean SAC already avoids the accumulation of errors?
> >
> > > **For cheetah-run and humanoid-stand, it is possible that their sensitivity to bias causes the initial bias to slow down learning. Nevertheless, due to the annealing mechanism, AIQL achieves similar final scores to SAC in these tasks.**
> >
> > - I'm a bit confused by such explanation. If it is actually the case that the initial bias has a strong influence, then why the proposed method, with the main motivation to deal with bias, cannot out-perform SAC?

---

> > > ### Author Response · Authors · 2024-11-25
> > > **Additional Response (1/1)**
> > >
> > > Thank you for your response! Below are our replies to those concerns.
> > >
> > > > - **Motivated by such over-simplified formulation, as suggested in the references, solutions could help adjust bias. But that does not mean the theoretical analysis itself has enough capability in modelling the actual settings. Still, it's good to know the current literature with similar approaches.**
> > > > - **Seems that the authors agree that no theoretical analysis has been provided for why such biases accumulate in the training.**
> > >
> > > If deriving the bias is not necessary, the occurrence of overestimation can be easily demonstrated. In Q-learning, when $Q$ is noisy due to function approximation or other factors, the true target is:
> > > $$ r + \max_{a'} \mathbb{E}_{Q}[Q(s', a')], $$
> > >
> > > but the target actually computed is:
> > > $$ \mathbb{E}_{Q}[r + \max _{a'} Q(s', a')]. $$
> > >
> > > According to Jensen's inequality, this leads to:
> > >
> > > $$ r + \max _{a'} \mathbb{E} _{Q} [Q(s', a')] \leq \mathbb{E} _{Q} [r + \max _{a'} Q(s', a')] $$
> > >
> > > , which results in the occurrence of overestimation bias.
> > >
> > > Let $Q'$ denote the Q-function with this positive bias added (such that $ Q'(s, a) \geq Q(s, a) $ for any $s,a$). Using $Q'$, the target becomes:
> > > $ \mathbb{E}[r + \max_{a'} Q'(s', a')] \geq \mathbb{E}[r + \max_{a'} Q(s', a')] \geq r + \max_{a'} \mathbb{E}_Q[Q(s', a')].$ This shows that the bias accumulates during the learning process.
> > >
> > > Here, no assumptions are made about the independence or distribution of $Q$. The accumulation of bias during the learning process is also discussed in XQL [1], which, using extreme value theory, demonstrates that the distribution of $Q$ evolves into a skewed distribution with a positive bias (assuming independence).
> > >
> > >
> > >
> > > > **According to Figure 7., SAC performs the best in most training steps. Does it mean SAC already avoids the accumulation of errors?**
> > >
> > > In SAC , instead of computing the maximum, the expected value $r + \mathbb{E}_{a' \sim \pi}[Q(s', a')]$ is calculated, making the occurrence of bias less likely (though not entirely absent, as bias can arise through the policy, as shown in [2]). Our goal is not to reduce the bias compared to SAC but to perform policy improvement using the maximum-like calculation by the IQL loss while preventing performance degradation from the accumulation of bias in the later stages of training through annealing.
> > >
> > > In our method, since the maximum-like calculation is performed, bias is larger than in SAC. However, policy improvement during the critic’s training enhances learning efficiency ([1, 3] are similarly motivated). Figure 7 demonstrates that when the expectile parameter $\tau$ is fixed, bias and skewness accumulate over time. While it is true that SAC exhibits the smallest bias and skewness, it also lacks policy improvement in the critic training process.
> > >
> > > Moreover, overestimation bias has been experimentally shown to promote exploration [4]. In AIQL, bias is suppressed in the later stages of training, allowing for both policy improvement through a maximum calculation in the earlier stage and an adjustment between exploration and exploitation, similar to an epsilon-greedy policy.
> > >
> > >
> > > > **I'm a bit confused by such explanation. If it is actually the case that the initial bias has a strong influence, then why the proposed method, with the main motivation to deal with bias, cannot out-perform SAC?**
> > >
> > > As mentioned above, in the early stages of training, our method  promotes policy improvement and exploration while increasing bias. In tasks where the performance is highly sensitive to bias and the benefits of policy improvement or exploration are limited, it is possible that our approach could yield worse results compared to SAC, which has smaller bias.
> > >
> > > Although time is limited, please feel free to ask if you have any concerns or questions!
> > >
> > >
> > > **References**
> > >
> > > [1] Divyansh Garg, Joey Hejna, Matthieu Geist, and Stefano Ermon. Extreme q-learning: Maxent RL without entropy. In International Conference on Learning Representations, 2023.
> > >
> > > [2] Scott Fujimoto, Herke van Hoof, and David Meger. Addressing function approximation error in actor-critic methods. In Jennifer Dy and Andreas Krause (eds.), Proceedings of the 35th International Conference on Machine Learning, volume 80 of Proceedings of Machine Learning Research, pp. 1587–1596. PMLR, 10–15 Jul 2018.
> > >
> > > [3] Ilya Kostrikov, Ashvin Nair, and Sergey Levine. Offline reinforcement learning with implicit qlearning. In International Conference on Learning Representations, 2022.
> > >
> > > [4] Qingfeng Lan, Yangchen Pan, Alona Fyshe, and Martha White. Maxmin q-learning: Controlling the estimation bias of q-learning. In International Conference on Learning Representations, 2020.

---

> > > > ### Author Response · Authors · 2024-11-28
> > > >
> > > > To provide better clarity on the motivation and key points of the paper, we have included additional explanations in the Introduction and Section 3.3. We would greatly appreciate it if you could review these changes along with our earlier response and share your thoughts.

---

> ### Author Response · Authors · 2024-11-21
> **Response (2/2)**
>
> **Q7. Why is there no comparison with some online version of IQL, if any?**
>
> In IQL, the policy is trained to avoid deviating significantly from the dataset policy, making it less suitable for direct use in online RL. As a result, an alternative training method is required. To address this, we implemented our approach by replacing only the critic's loss in SAC, which is widely used in online RL. In the additional experiments combining our method with TD3, where entropy maximization is absent, the critic's learning closely resembles that of IQL in offline RL.
>
>
> **References**
>
> [1] Sebastian Thrun and Anton Schwartz. Issues in using function approximation for reinforcement learning. In Fourth Connectionist Models Summer School, 1993.
>
> [2] Qingfeng Lan, Yangchen Pan, Alona Fyshe, and Martha White. Maxmin q-learning: Controlling the estimation bias of q-learning. In International Conference on Learning Representations, 2020.
>
> [3] Xinyue Chen, Che Wang, Zijian Zhou, and Keith W. Ross. Randomized ensembled double q-learning: Learning fast without a model. In International Conference on Learning Representations, 2021.

---

> ### Comment · Reviewer_9sVB · 2024-11-28
>
> Thanks for the additional details.
>
> > **Proof for accumulation of over-estimation bias**
>
> - For convenience of reference, let's refer to the equations in your last reply as 'Equation 1-3', and the last inequality following the sentence 'the target becomes' as 'the inequality'. Then according the reviewer's understanding, through the process of online learning, the policy is being changed. Then the estimation bias, in terms of Q-function, should be the difference between the estimated Q of one specific intermediate policy and its underlying Q-function. Now going back to the derivation in the reply: by Equation 3, one can argue that the estimation of Q, in the sense of empirical loss function, is performed with a possibly higher target. But this does not guarantee that Q can be over-estimated for all state-action pairs simultaneously. In addition, when talking about the estimation bias due to $Q\^\\prime$, it is with respect to the policy in the next iteration, which is different from the one concerned when talking about using $Q$ as the target. That means one need to compare $\\hat{Q}\^{\\pi_{i+1}} - Q\^{\\pi_{i+1}}$ and $\\hat{Q}\^{\\pi_{i}} - Q\^{\\pi_{i}}$, which is not determined by the inequality provided. If I misunderstood the problem setting, please let me know.
>
> > **Comparison**
>
> - The clarification about the implementation-performance-wise difference between the method and SAC looks reasonable.

---

> ### Author Response · Authors · 2024-12-02
>
> Thank you for your response!
>
> >**Proof for accumulation of over-estimation bias**
>
> First, in value function learning where the max operator is applied, as in Q-learning, this process corresponds to value iteration rather than policy iteration.  Therefore, it does not estimate the value function of a specific policy but instead performs Bellman optimality backups to approximate the optimal Q-value (Q*). Consequently, evaluating $ \hat{Q}^{\pi_k} - {Q}^{\pi_k}$ is not feasible. Furthermore, it is challenging to accurately calculate and compare the bias at each update step in a general form.  Given that the discount rate contributes to bias reduction, the use of the term "accumulation" may require reconsideration, and we will revise such expressions accordingly. However, the fact that max leaves residual bias is evident from the left plot in Figure 7, where the bias increases at the final steps as $\tau$ grows. Furthermore, due to the inherent randomness in training methods like stochastic gradient descent, the Q-function remains noisy over time. This noise, based on Jensen's inequality, continuously introduces bias. This suggests that convergence to an unbiased value is hindered, and the annealing of $\tau$ is effective in mitigating this issue. We will revise the relevant expressions in the paper for the camera-ready version to clarify these points.
>
>
> We would greatly appreciate any additional feedback you might have, as the rebuttal period will be concluding soon.

---

### Official Review · Reviewer_Vdto · 2024-11-04

**Soundness:** 3
**Presentation:** 2
**Contribution:** 2
**Rating:** 3
**Confidence:** 3

**Summary:**

This paper uses an annealed implicit Q-learning loss in online RL with SAC, to replace the standard MSE loss. The loss is annealed in a way that increases bias in the early stage of training and reduce bias toward the later stage.

**Strengths:**

- The paper proposes a very simple trick that can potentially improve the sample efficiency of training the Q-function in SAC.
- The experiments show a major improvement in hopper-hop and humanoid-run environments.
- The paper provides a large set of results on various $\tau$ values being annealed or kept fixed.

**Weaknesses:**

## Lacks justification of expectile loss

While the paper spends a significant portion trying to motivate the $\tau$ schedule, it is still unclear why introduce the implicit Q-learning loss to begin with? What benefit does it offer over the thoeretically guaranteed objective of SAC in online RL?

In other words, it is unclear why AIQL outperforms SAC in the environments considered. Figure 7 shows that the skewness and bias of SAC are the minimum, and lower than AIQL. Then what is the root cause of AIQL outperforming SAC?


## The annealing schedule of optimality is not sufficiently justified
I am still unable to understand why the bias should be kept high in the beginning (and lower in the end)?
Section 3.2 discusses overestimation bias and that too much overestimation bias would lead to failure in learning, which makes sense.
Section 3.3 says "$\tau$ control the trade-off between optimality and bias", but this relationship was never justified. Therefore, the connection between skewness/bias and $\tau$ value is not well established.


## The scheduling of tau depends heavily on the environment
Essentially this paper requires two kinds of hyperparameters:
- the start and end values of $\tau$.
- the total duration of learning (set to 3M steps in this work) to set up the linear schedule.

But how does one select the decay schedule for an environment where the schedule is not known a priori? Even for the humanoid tasks, the convergence is not fully achieved, e.g., humanoid-run should reach max return of ~450 if trained for longer. So, how does one define the annealing schedule in such cases?

I would encourage the authors to demonstrate their idea on more diverse environments apart from DM Control, to fully understand when the expectile loss and the annealing schedule can help or hurt. And also how to set the hyperparameters in general.

## IQL loss on algorithms other than SAC
The paper claims "tested the modeling of optimal Q-value using implicit Q-learning loss in continuous-action online RL", but only shows results on SAC. Would similar improvements be expected in TD3 as well? The authors already have the TD3 baseline, so I expect this experiment to not be so hard as the incorporation of expectile loss should be pretty simple?

## Interpretation of the values of $\tau$ annealing
In Section 4.3, Annealed (0.7) and Fixed (0.7) perform quite well, on par with Annealed (0.9). I am not sure whether there is a strong justification for the paper's insight that one needs to have a high bias in the beginning. In other words, I am still not sure where the learning gains come from.

Similarly, Fixed (0.6) does really well (713, 822) but Fixed (0.5), i.e. SAC, underperforms quite a bit (657, 765). What happens between 0.5 and 0.6 that causes the improvement? I believe investigating this could give a clear insight about why expectile loss is helping, which this paper is currently missing.

**Questions:**

1. The paper mentions:
> As learning progresses, the policy approaches the optimal policy, reducing the necessity for high optimality in value function learning.

Why is high optimality not necessary towards the end? If the value function is suboptimal, then the policy would also be suboptimal.

2. The paper categorizes hopper-hop and humanoid-run as "more challenging tasks". What makes hopper-hop more challenging than quadruped-run?

3. Replacing SAC's value function update with expectile loss does mean that the entropy regularization is removed, correct?

4. What is the reason for AIQL to perform so well in hopper-hop? It seems this is the main reason for the cumulative results to be so good.

---

> ### Author Response · Authors · 2024-11-21
> **Response (1/2)**
>
> Thank you for your valuable comments and questions! We would like to address the concerns raised under Weaknesses and Questions.
>
> **Q1. Lacks justification of expectile loss**
>
>  > it is still unclear why introduce the implicit Q-learning loss to begin with?
>
> The primary motivation for introducing the IQL loss lies in enabling the calculation of maxQ(s′,a′) during the critic's update process. In actor-critic methods for continuous action tasks, such as SAC and TD3, policy improvement depends solely on the actor. Our study aims to enhance performance by enabling improvement through the critic as well, achieved by calculating the maximum value. This motivation is consistent with that stated in XQL, one of the comparison methods in our research, and is similarly rooted in prior work like Soft Q-Learning, which also computes a (soft) maximum. However, XQL suffers from instability, and Soft Q-Learning requires complex approximate inference procedures, prompting the development of SAC. In contrast, we used IQL loss as a stable alternative to XQL loss, which involves exponential terms, that implicitly calculates the maximum from the samples with simplicity.
>
> By applying this loss to SAC, we aimed to demonstrate performance improvements by merely replacing the loss function of a commonly used method. The choice of SAC as the base algorithm is not obligatory. To further validate this, we added experiments combining AIQL with TD3 in Section D, where similar performance improvements were observed.
>
> > Then what is the root cause of AIQL outperforming SAC?
>
> Therefore, in response to the question of why AIQL performs better than SAC, we believe this is due to more efficient policy improvement as well as the influence of bias. As you pointed out, SAC exhibits lower bias and skewness compared to AIQL. However, prior work such as [1] has experimentally demonstrated that positive bias can effectively encourage exploration.
>
> From Figure 7, we observe that AIQL has a larger bias during the early stages of training, which gradually diminishes to levels comparable to SAC in the later stages. This behavior resembles an epsilon-greedy policy, where exploration is emphasized during the early stages of learning, while exploitation dominates in the later stages, facilitated by annealing.
>
> We believe that this annealing mechanism and exploration driven by bias significantly contribute to the performance improvements seen in AIQL.
>
> **Q2. The annealing schedule of optimality is not sufficiently justified**
> >I am still unable to understand why the bias should be kept high in the beginning (and lower in the end)?
>
> As stated earlier in Section 3.2, overestimation bias arises from the calculation of max⁡Q(s′,a′)  [2]. With the expectile loss, the closer $\tau$ is to 1, the closer it approximates the maximum, whereas at $\tau = 0.5$, it calculates the expectation.
>
> This means that as $\tau$ approaches 1, overestimation bias becomes more likely to occur, while it becomes less likely as $\tau$ approaches 0.5. At the same time, when $\tau$ is close to 1, the loss approximates the optimal value through a maximum-like calculation, whereas at $\tau = 0.5$, it estimates the value function of the policy.
>
> Based on this understanding, we claim that $\tau$ controls the trade-off between optimality and bias. We have added this explanation in Section 3.3 for clarity.
>
> **Q3. The scheduling of tau depends heavily on the environment**
>
> > So, how does one define the annealing schedule in such cases?
>
> The end value of $\tau$ could also be treated as a hyperparameter, but we recommend setting it to 0.5. By doing so, the algorithm transitions into SAC at the end of annealing, allowing the learning process to continue as SAC afterward. While the start value of $\tau$ and the annealing duration are hyperparameters, they introduce a continuous change rather than a discrete one to SAC. For example, setting the start value of $\tau$ to 0.5 and the annealing duration to 0 results in SAC. By gradually increasing the start value of $\tau$ and the annealing duration, the benefits of our method can be realized.
>
> Table 3 demonstrates that the algorithm is robust to the specific values of $\tau$ when annealing is employed. Additionally, we have added experimental results regarding the duration of annealing in Section E. Even with an annealing duration as short as just 1 million steps, our method significantly outperforms SAC. Furthermore, the algorithm is not sensitive to the annealing duration, making it practical to use.
>
> **Q4. IQL loss on algorithms other than SAC**
>
> We have added results in Section D combining AIQL with TD3 for the tasks where AIQL with SAC had the most significant impact — hopper-hop and humanoid-run. Similar to the results of AIQL+SAC, we observed significant performance improvements. Experiments on other tasks will be included in the camera-ready version.

---

> ### Author Response · Authors · 2024-11-21
> **Response (2/2)**
>
> **Q5. Interpretation of the values of τ annealing**
>
> > What happens between 0.5 and 0.6 that causes the improvement?
>
> Regarding the difference between Fixed (0.6) and SAC, we believe it stems from whether or not there is even a slight occurrence of policy improvement and bias during the critic's learning process, as discussed in response to A1. Improvements and biases accumulate as learning progresses, so even a small increase in $\tau$ from 0.5 should have an impact.
>
> In Figure 7, it is evident that Fixed (0.6) exhibits greater skewness and bias compared to SAC. Similarly, in the IQL paper, although $\tau = 0.7$ (not so close to 1) is often used across various tasks, the performance significantly surpasses that of AWAC, which is similar to IQL with $\tau=0.5$.
>
> **Q6. Why is high optimality not necessary towards the end? If the value function is suboptimal, then the policy would also be suboptimal.**
>
> We had assumed that an optimal policy and value function could be learned; however, there is also the possibility that they may remain suboptimal. Our intention was to convey a state in which further policy improvement is no longer achievable. We have revised the relevant statement in Section 3.3 to make it more precise.
>
> **Q7. The paper categorizes hopper-hop and humanoid-run as "more challenging tasks". What makes hopper-hop more challenging than quadruped-run?**
>
> We assessed the difficulty of tasks based on the performance of prior studies. In the DM Control suite, the rewards are normalized to 1 per step, and each episode consists of 1000 steps, resulting in a maximum possible return of 1000. For the quadruped-run task, returns exceeding 800 have been achieved with methods like SAC and TD3. However, for tasks like hopper-hop and humanoid-run, returns are approximately 100-200, indicating that these are more challenging tasks.
>
> **Q8. Replacing SAC's value function update with expectile loss does mean that the entropy regularization is removed, correct?**
>
> We did not remove entropy regularization. Instead, we simply replaced the L2 loss in the critic's loss function with expectile loss, without altering the rest of the loss function.
>
> **Q9. What is the reason for AIQL to perform so well in hopper-hop? It seems this is the main reason for the cumulative results to be so good.**
>
> The improvement in AIQL's performance, as mentioned in response to Q1, is attributed to the efficiency of policy improvement and the exploration facilitated by bias. One way to enhance policy improvement is by increasing the number of updates (replay ratio) at the expense of computational cost. According to Appendix D.1 of [3], increasing the replay ratio significantly improves the performance in the hopper-hop task. This suggests that, in hopper-hop, policy improvement has a substantial impact on improving sample efficiency.
>
> **References**
>
> [1]  Qingfeng Lan, Yangchen Pan, Alona Fyshe, and Martha White. Maxmin q-learning: Controlling the estimation bias of q-learning. In International Conference on Learning Representations, 2020.
>
> [2] Sebastian Thrun and Anton Schwartz. Issues in using function approximation for reinforcement learning. In Fourth Connectionist Models Summer School, 1993.
>
> [3] P. D’Oro, M. Schwarzer, E. Nikishin, P.-L. Bacon, M. G. Bellemare, and A. Courville. Sampleefficient reinforcement learning by breaking the replay ratio barrier. In International Conference on Learning Representations, 2023.

---

> > ### Author Response · Authors · 2024-11-28
> >
> > To strengthen the claims of our paper, we have conducted additional experiments, which are now included in Appendix D, E, and F. Furthermore, to clarify the motivation and main points of the paper, we have added detailed explanations in the Introduction and Section 3.3. We kindly request that you review these updates, along with our previous response, and share your feedback with us.

---

> > > ### Comment · Reviewer_Vdto · 2024-11-30
> > >
> > > My primary concern is still about why IQL in online RL makes sense.
> > >
> > > > The primary motivation for introducing the IQL loss lies in enabling the calculation of maxQ(s′,a′) during the critic's update process. In actor-critic methods for continuous action tasks, such as SAC and TD3, policy improvement depends solely on the actor. Our study aims to enhance performance by enabling improvement through the critic as well, achieved by calculating the maximum value.
> > >
> > > I believe the IQL paper already has experiments on online RL fine-tuning with IQL, so I don't see how that can be claimed as a novelty for this work. Is my understanding incorrect about IQL's results? If there is something more unique about this paper's results pertaining to online RL, this work should show theoretically why IQL in online RL makes more sense than SAC alone. As of now, it seems like a heuristic that happens to work on certain environments.
> > >
> > > Furthermore, the performance of SAC and TD3 seem to be truncated and exceptionally low in these environments. Are these methods really fully hyperparameter-tuned to be state-of-the-art?
> > >
> > > The gains coming primarily from hopper and humanoid seems unconvincing to me and there is no direct evidence that incorporating the "max" step in IQL is the direct cause of this improvement in these environments.
> > >
> > > There needs to be a better demonstration of why exactly IQL works, more than that "it happens to work" in these environments.
> > > Also, I think the authors should report results on a different set of benchmarks than dm control to really justify this difference empirically.

---

> ### Author Response · Authors · 2024-12-02
>
> Thank you for the additional feedback. Below is our responses:
>
> >**I believe the IQL paper already has experiments on online RL fine-tuning with IQL, so I don't see how that can be claimed as a novelty for this work. Is my understanding incorrect about IQL's results?**
>
> While fine-tuning is performed online after offline training in IQL paper, to the best of our knowledge, there is no existing work that trains an agent from scratch in an online setting using the IQL loss and compares it with widely-used online RL methods. We consider it a significant finding that a simple modification to the critic loss in existing methods can lead to performance improvements. Furthermore, the annealing of the $\tau$ parameter, implemented to suppress the increased bias introduced by using the IQL loss, is an approach uniquely feasible in the context of online RL. This approach enhances performance and makes the method more robust to the choice of $\tau$, which we regard as another key contribution of our work.
>
> >**Furthermore, the performance of SAC and TD3 seem to be truncated and exceptionally low in these environments. Are these methods really fully hyperparameter-tuned to be state-of-the-art?**
>
> For the SAC implementation, we use the same hyperparameters as the original paper, while for TD3, we rely on the authors' implementation. We use the same hyperparameters when combining them with AIQL. Regarding the training steps, we extended the training to 5 million steps and added the results to Figure 13, which similarly show that the proposed method outperforms SAC. Moreover, the performance improvements in shorter training steps are also valuable from the perspective of improving sample efficiency.
>
> >**The gains coming primarily from hopper and humanoid seems unconvincing to me and there is no direct evidence that incorporating the "max" step in IQL is the direct cause of this improvement in these environments.**
>
> In tasks other than Hopper and Humanoid, SAC's returns are close to 800, leaving limited room for improvement. However, even in tasks like Quadruped and Walker, AIQL demonstrates better performance than SAC, suggesting that AIQL is consistently effective. Furthermore, the utility of the "max" operation is evidenced by the experiments on max-backup in Figure 6 of our paper. In max-backup, the only modification to SAC is in the computation of the target value for the critic update, where multiple actions are sampled, and the maximum action is selected. Comparing this with the SAC returns shown in Figure 4, max-backup achieves higher returns. We believe this demonstrates the value of introducing the "max" step to accelerate policy improvement, which is a key element in RL.
>
> >**There needs to be a better demonstration of why exactly IQL works, more than that "it happens to work" in these environments. Also, I think the authors should report results on a different set of benchmarks than dm control to really justify this difference empirically.**
>
> While we believe that evaluations on various locomotion tasks in DM Control demonstrate the effectiveness of our method, we are also conducting additional experiments on manipulation tasks in [1]. Due to the limited time and computational resources, we are uncertain whether we can share the results before the end of the rebuttal period. Nevertheless, we will share them as soon as they are completed.
>
> [1] T. Yu, D. Quillen, Z. He, R. Julian, K. Hausman, C. Finn, and S. Levine. Meta-world: A benchmark and evaluation for multi-task and meta reinforcement learning. In Conference on Robot Learning (CoRL), 2020.

---

> > ### Comment · Reviewer_Vdto · 2024-12-02
> >
> > > While fine-tuning is performed online after offline training in IQL paper, to the best of our knowledge, there is no existing work that trains an agent from scratch in an online setting using the IQL loss and compares it with widely-used online RL methods
> >
> > In this case, the paper's claims should be adjusted accordingly to focus on learning from "scratch". Currently, the paper's main contribution is: "We investigated and experimentally tested the modeling of optimal Q-value using implicit Q-learning loss in continuous-action online RL." This is not consistent with what IQL paper already explored.
> >
> > Since the annealing requirement comes explicitly due to the constraint of "learning from scratch", there needs to be an equivalent analysis on fine-tuning like IQL did. What happens there — would annealing help there too or is it not necessary? What is the fundamental difference between learning from scratch and fine-tuning that motivates the requirement of annealing?
> >
> > I think correcting the claimed contribution would require adjustments to the paper to be centered around that contribution.
> >
> > > There needs to be a better demonstration of why exactly IQL works, more than that "it happens to work" in these environments. Also, I think the authors should report results on a different set of benchmarks than dm control to really justify this difference empirically.
> >
> > While I understand my comment requires more experiments and there is not much time left in the rebuttal, so I do not expect the authors to conduct these experiments in the time left. I have assessed the paper with the information I have access to, and do not feel convinced to revise the scores, especially due to the questions on experiment viability and generalizability to other benchmarks, lack of theoretical justification of modifying the policy evaluation step of GPI, and the discrepancy in the claimed contribution and what prior IQL did.
> >
> > I sincerely believe this idea of annealing $\tau$ can be meaningful and potentially lead to improvements in RL. It would also throw more light on the practical importance of expected SARSA and max Q-learning in deep RL. However, the current state of the paper does not do justice to this. I hope the authors can revise the paper to make their claims stronger and more robust to doubts.

---

> > > ### Author Response · Authors · 2024-12-04
> > >
> > > Thank you for your valuable feedback.
> > >
> > > The fine-tuning experiments in the IQL paper do not undermine the contributions of this work. Fine-tuning in IQL is conducted exclusively after offline training within IQL and employs comparison methods specifically tailored to offline RL and fine-tuning, thereby verifying its effectiveness in very limited scenarios. In contrast, our study focuses on general online RL from scratch, which fundamentally differs in scope. While annealing is irrelevant in offline IQL as only the final value function influences the policy, it could be effective in fine-tuning. However, like many other studies on online RL from scratch, our research does not focus on fine-tuning from offline RL. We will add a clarification to our paper that fine-tuning approaches, such as those in the IQL paper, are outside the scope of this study.
> > >
> > >
> > > Regarding experiment viability, the code for our experiments has been uploaded. To provide benchmarks beyond DM Control’s locomotion tasks, we conducted experiments on two manipulation tasks in Meta-World (not multi-task): pick-place and dial-turn. In the pick-place task, AIQL constantly outperformed SAC within the learning range. For the dial-turn task, although all methods have already saturated in performance, AIQL demonstrated better performance with fewer steps compared to SAC, indicating improved sample efficiency. These results suggest the generalizability of AIQL to other benchmarks and further strengthen the argument for the effectiveness of max-based updates in the critic. We will also include results from other tasks.
> > >
> > >
> > > - pick-place
> > > |  | 0.5M steps | 1.0M steps | 1.5M steps | 2.0M steps | 2.5M steps |
> > > | :--- | ---: | ---: | ---: | ---: | ---: |
> > > | SAC | 13 | 140 | 621 | 969 | 1176 |
> > > | AIQL ($\tau=0.6$) | 31 | 624 | 1213 | 1754 | 2635 |
> > > | AIQL ($\tau=0.7$) | 64 | 452 | 1225 | 2144 | 2978 |
> > > | AIQL ($\tau=0.8$) | 14 | 129 | 636 | 1271 | 2053 |
> > > | AIQL ($\tau=0.9$) | 17 | 53 | 678 | 1072 | 1879 |
> > >
> > >
> > > - dual-turn
> > > |  | 0.5M steps | 1.0M steps | 1.5M steps |
> > > | :--- | ---: | ---: | ---: |
> > > | SAC | 2909 | 3892 | 4235 |
> > > | AIQL ($\tau=0.6$) | 3748 | 4208 | 4302 |
> > > | AIQL ($\tau=0.7$) | 3844 | 4139 | 4237 |
> > > | AIQL ($\tau=0.8$) | 3428 | 3880 | 4259 |
> > > | AIQL ($\tau=0.9$) | 3670 | 4230 | 4205 |

---

### Meta-Review · Area_Chair_w5Rd · 2024-12-20

**Metareview:**

This paper proposes AIQL (Annealed Implicit Q-learning), a new method for improving sample efficiency in online reinforcement learning with continuous action spaces. AIQL builds on SAC (Soft Actor-Critic) by incorporating an annealed implicit Q-learning loss, which gradually reduces optimality bias during training.

Strengths
-----------
- **Simple and effective:** AIQL introduces a relatively simple modification to SAC, yet demonstrates significant performance gains in certain environments, particularly Hopper-Hop and Humanoid-Run.

- **Thorough evaluation:** The paper provides a comprehensive set of results, exploring different annealing schedules and hyperparameter settings.

Weaknesses
---------------
- **Limited motivation:** The motivation for using the implicit Q-learning loss and the specific annealing schedule remains unclear. The paper lacks a strong theoretical or empirical justification for these choices.

- **Lack of generalization:** The improvements are primarily demonstrated on DM Control tasks. It remains to be seen whether AIQL generalizes to other environments and RL algorithms beyond SAC.

- **Hyperparameter sensitivity:** The performance of AIQL depends on the annealing schedule, which in turn relies on knowing the total training duration. This limits its applicability to new environments where the optimal schedule is unknown.

- **Limited novelty:** Some reviewers found the contribution incremental, as it primarily applies existing techniques (IQL and annealing) to a new setting.

AIQL presents a promising approach for improving sample efficiency in online continuous-action RL. However, the paper needs further development to address the concerns regarding justification, generalization, and hyperparameter sensitivity. In particular, it would be desirable to have a stronger theoretical or empirical justification for using the implicit Q-learning loss and the annealing schedule.
Evaluate AIQL on a wider range of environments and RL algorithms (e.g., TD3, CrossQ).

**Additional Comments On Reviewer Discussion:**

The discussion has been mainly focused on the relevance of the contribution and especially on its motivation. Despite the authors' rebuttal, some reviewers' concerns remained unsolved.

---

### Decision · Program_Chairs · 2025-01-22

Reject